# COLORIZATION TRANSFORMER

**Manoj Kumar, Dirk Weissenborn & Nal Kalchbrenner**
Google Research, Brain Team
{mechcoder,diwe,nalk}@google.com

## ABSTRACT

We present the Colorization Transformer, a novel approach for diverse high fidelity image colorization based on self-attention. Given a grayscale image, the colorization proceeds in three steps. We first use a conditional autoregressive transformer to produce a low resolution coarse coloring of the grayscale image. Our architecture adopts conditional transformer layers to effectively condition grayscale input. Two subsequent fully parallel networks upsample the coarse colored low resolution image into a finely colored high resolution image. Sampling from the Colorization Transformer produces diverse colorings whose fidelity outperforms the previous state-of-the-art on colorising ImageNet based on FID results and based on a human evaluation in a Mechanical Turk test. Remarkably, in more than 60% of cases human evaluators prefer the highest rated among three generated colorings over the ground truth. The code and pre-trained checkpoints for Colorization Transformer are publicly available at this url.

## 1 INTRODUCTION

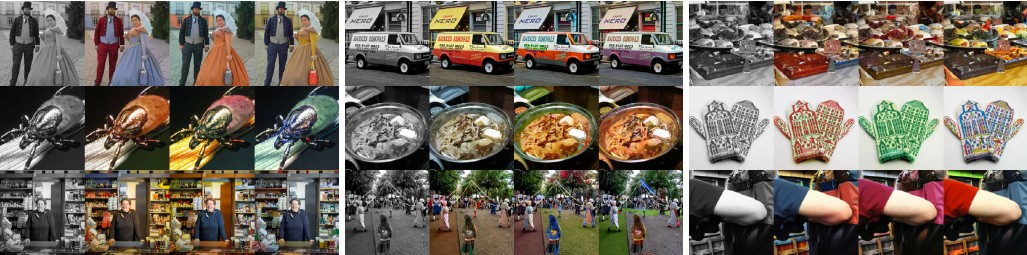

**Figure 1:** Samples of our model showing diverse, high-fidelity colorizations.

Image colorization is a challenging, inherently stochastic task that requires a semantic understanding of the scene as well as knowledge of the world. Core immediate applications of the technique include producing organic new colorizations of existing image and video content as well as giving life to originally grayscale media, such as old archival images (Tsaftaris et al., 2014), videos (Geshwind, 1986) and black-and-white cartoons (Sỳkora et al., 2004; Qu et al., 2006; Cinarel & Zhang, 2017). Colorization also has important technical uses as a way to learn meaningful representations without explicit supervision (Zhang et al., 2016; Larsson et al., 2016; Vondrick et al., 2018) or as an unsupervised data augmentation technique, whereby diverse semantics-preserving colorizations of labelled images are produced with a colorization model trained on a potentially much larger set of unlabelled images.

The current state-of-the-art in automated colorization are neural generative approaches based on log-likelihood estimation (Guadarrama et al., 2017; Royer et al., 2017; Ardizzone et al., 2019). Probabilistic models are a natural fit for the one-to-many task of image colorization and obtain better results than earlier determinisitic approaches avoiding some of the persistent pitfalls (Zhang et al., 2016). Probabilistic models also have the central advantage of producing multiple diverse colorings that are sampled from the learnt distribution.

In this paper, we introduce the Colorization Transformer (ColTran), a probabilistic colorization model composed only of axial self-attention blocks (Ho et al., 2019b; Wang et al., 2020). The main

advantages of axial self-attention blocks are the ability to capture a global receptive field with only two layers and $\mathcal{O}(D\sqrt{D})$ instead of $\mathcal{O}(D^2)$ complexity. They can be implemented efficiently using matrix-multiplications on modern accelerators such as TPUs (Jouppi et al., 2017). In order to enable colorization of high-resolution grayscale images, we decompose the task into three simpler sequential subtasks: coarse low resolution autoregressive colorization, parallel color and spatial super-resolution. For coarse low resolution colorization, we apply a conditional variant of Axial Transformer (Ho et al., 2019b), a state-of-the-art autoregressive image generation model that does not require custom kernels (Child et al., 2019). While Axial Transformers support conditioning by biasing the input, we find that directly conditioning the transformer layers can improve results significantly. Finally, by leveraging the semi-parallel sampling mechanism of Axial Transformers we are able to colorize images faster at higher resolution than previous work (Guadarrama et al., 2017) and as an effect this results in improved colorization fidelity. Finally, we employ fast parallel deterministic upsampling models to super-resolve the coarsely colorized image into the final high resolution output. In summary, our main contributions are:

- First application of transformers for high-resolution ($256 \times 256$) image colorization.
- We introduce conditional transformer layers for low-resolution coarse colorization in Section 4.1. The conditional layers incorporate conditioning information via multiple learnable components that are applied per-pixel and per-channel. We validate the contribution of each component with extensive experimentation and ablation studies.
- We propose training an auxiliary parallel prediction model jointly with the low resolution coarse colorization model in Section 4.2. Improved FID scores demonstrate the usefulness of this auxiliary model.
- We establish a new state-of-the-art on image colorization outperforming prior methods by a large margin on FID scores and a 2-Alternative Forced Choice (2AFC) Mechanical Turk test. Remarkably, in more than 60% of cases human evaluators prefer the highest rated among three generated colorings over the ground truth.

## 2 RELATED WORK

Colorization methods have initially relied on human-in-the-loop approaches to provide hints in the form of scribbles (Levin et al., 2004; Ironi et al., 2005; Huang et al., 2005; Yatziv & Sapiro, 2006; Qu et al., 2006; Luan et al., 2007; Tsaftaris et al., 2014; Zhang et al., 2017; Ci et al., 2018) and exemplar-based techniques that involve identifying a reference source image to copy colors from (Reinhard et al., 2001; Welsh et al., 2002; Tai et al., 2005; Ironi et al., 2005; Pitié et al., 2007; Morimoto et al., 2009; Gupta et al., 2012; Xiao et al., 2020). Exemplar based techniques have been recently extended to video as well (Zhang et al., 2019a). In the past few years, the focus has moved on to more automated, neural colorization methods. The deterministic colorization techniques such as CIC (Zhang et al., 2016), LRAC (Larsson et al., 2016), LTBC (Iizuka et al., 2016), Pix2Pix (Isola et al., 2017) and DC (Cheng et al., 2015; Dahl, 2016) involve variations of CNNs to model per-pixel color information conditioned on the intensity.

Generative colorization models typically extend unconditional image generation models to incorporate conditioning information from a grayscale image. Specifically, cINN (Ardizzone et al., 2019) use conditional normalizing flows (Dinh et al., 2014), VAE-MDN (Deshpande et al., 2017; 2015) and SCC-DC (Messaoud et al., 2018) use conditional VAEs (Kingma & Welling, 2013), and cGAN (Cao et al., 2017) use GANs (Goodfellow et al., 2014) for generative colorization. Most closely related to ColTran are other autoregressive approaches such as PixColor (Guadarrama et al., 2017) and PIC (Royer et al., 2017) with PixColor obtaining slightly better results than PIC due to its CNN-based upsampling strategy. ColTran is similar to PixColor in the usage of an autoregressive model for low resolution colorization and parallel spatial upsampling. ColTran differs from PixColor in the following ways. We train ColTran in a completely unsupervised fashion, while the conditioning network in PixColor requires pre-training with an object detection network that provides substantial semantic information. PixColor relies on PixelCNN (Oord et al., 2016) that requires a large depth to model interactions between all pixels. ColTran relies on Axial Transformer (Ho et al., 2019b) and can model all interactions between pixels with just 2 layers. PixColor uses different architectures for conditioning, colorization and super-resolution, while ColTran is conceptually simpler as we use self-attention blocks everywhere for both colorization and superresolution. Finally, we train

our autoregressive model on a single coarse channel and a separate color upsampling network that improves fidelity (See: 5.3). The multi-stage generation process in ColTran that upsamples in depth and in size is related to that used in Subscale Pixel Networks (Menick & Kalchbrenner, 2018) for image generation, with differences in the order and representation of bits as well as in the use of fully parallel networks. The self-attention blocks that are the building blocks of ColTran were initially developed for machine translation (Vaswani et al., 2017), but are now widely used in a number of other applications including density estimation (Parmar et al., 2018; Child et al., 2019; Ho et al., 2019a; Weissenborn et al., 2019) and GANs (Zhang et al., 2019b)

## 3  BACKGROUND: AXIAL TRANSFORMER

### 3.1  ROW AND COLUMN SELF-ATTENTION

Self-attention (SA) has become a standard building block in many neural architectures. Although the complexity of self-attention is quadratic with the number of input elements (here pixels), it has become quite popular for image modeling recently (Parmar et al., 2018; Weissenborn et al., 2019) due to modeling innovations that don't require running global self-attention between all pixels. Following the work of (Ho et al., 2019b) we employ standard $\mathbf{qkv}$ self-attention (Vaswani et al., 2017) within rows and columns of an image. By alternating row- and column self-attention we effectively allow global exchange of information between all pixel positions. For the sake of brevity we omit the exact equations for multihead self-attention and refer the interested reader to the Appendix H for more details. Row/column attention layers are the core components of our model. We use them in the autoregressive colorizer, the spatial upsampler and the color upsampler.

### 3.2  AXIAL TRANSFORMER

Ths Axial Transformer (Ho et al., 2019b) is an autoregressive model that applies (masked) row- and column self-attention operations in a way that efficiently summarizes all past information $\mathbf{x}_{i,<j}$ and $\mathbf{x}_{<i,\cdot}$ to model a distribution over pixel $\mathbf{x}_{i,j}$ at position $i, j$. Causal masking is employed by setting all $A_{m,n} = 0$ where $n > m$ during self-attention (see Eq. 15).

**Outer decoder.**  The outer decoder computes a state $\mathbf{s}_o$ over all previous rows $\mathbf{x}_{\leq i,\cdot}$ by applying $N$ layers of full row self-attention followed by masked column self-attention. (Eq 2). $\mathbf{s}_o$ is shifted down by a single row, such that the output context $\mathbf{o}_{i,j}$ at position $i, j$ only contains information about pixels $\mathbf{x}_{<i,\cdot}$ from prior rows. (Eq 3)

$$\mathbf{e} = \text{Embeddings}(\mathbf{x}) \tag{1}$$
$$\mathbf{s}_o = \text{MaskedColumn}(\text{Row}(\mathbf{e})) \qquad \times N \tag{2}$$
$$\mathbf{o} = \text{ShiftDown}(\mathbf{s}_o) \tag{3}$$

**Inner decoder.**  The embeddings to the inner decoder are shifted right by a single column to mask the current pixel $\mathbf{x}_{i,j}$. The context $\mathbf{o}$ from the outer decoder conditions the inner decoder by biasing the shifted embeddings. It then computes a final state $\mathbf{h}$, by applying $N$ layers of masked row-wise self-attention to infuse additional information from prior pixels of the same row $\mathbf{x}_{i,<j}$ (Eq 4). $\mathbf{h}_{i,j}$ comprises information about all past pixels $\mathbf{x}_{<i}$ and $\mathbf{x}_{i,<j}$. A dense layer projects $\mathbf{h}$ into a distribution $p(\mathbf{x}_{ij})$ over the pixel at position $(i, j)$ conditioned on all previous pixels $\mathbf{x}_{i,<j}$ and $\mathbf{x}_{<i,\cdot}$.

$$\mathbf{z} = \mathbf{o} + \text{ShiftRight}(\mathbf{e}) \tag{4}$$
$$\mathbf{h} = \text{MaskedRow}(\mathbf{z}) \qquad \times N \tag{5}$$
$$p(\mathbf{x}_{ij}) = \text{Dense}(\mathbf{h}) \tag{6}$$

**Encoder.**  As shown above, the outer and inner decoder operate on 2-D inputs, such as a single channel of an image. For multi-channel RGB images, when modeling the "current channel", the Axial Transformer incorporates information from prior channels of an image (as per raster order) with an encoder. The encoder encodes each prior channel independently with a stack of unmasked row/column attention layers. The encoder outputs across all prior channels are summed to output a conditioning context $\mathbf{c}$ for the "current channel". The context conditions the outer and inner decoder by biasing the inputs in Eq 1 and Eq 4 respectively.

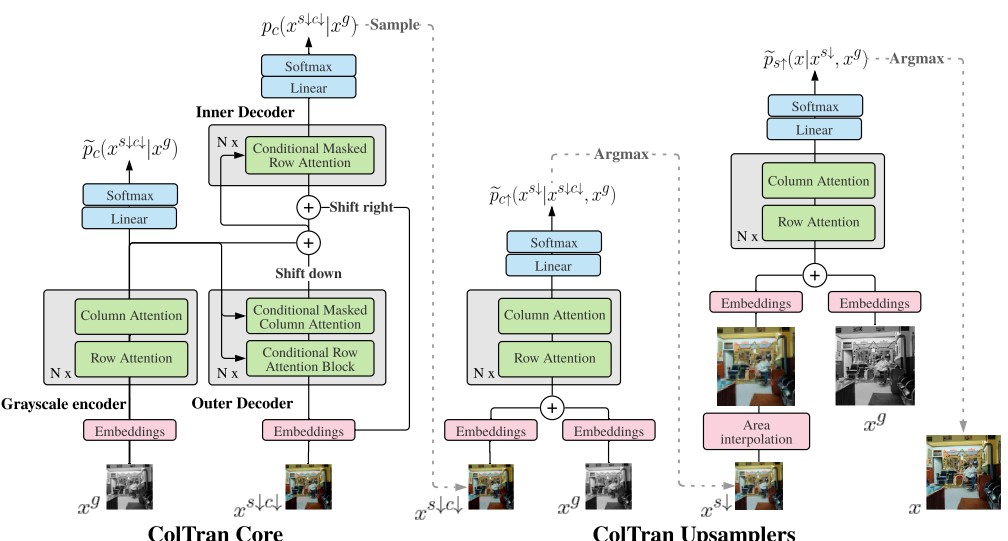

**Figure 2:** Depiction of ColTran. It consists of 3 individual models: an autoregressive colorizer (left), a color upsampler (middle) and a spatial upsampler (right). Each model is optimized independently. The autoregressive colorizer (ColTran core) is an instantiation of Axial Transformer (Sec. 3.2, Ho et al. (2019b)) with conditional transformer layers and an auxiliary parallel head proposed in this work (Sec. 4.1). During training, the ground-truth coarse low resolution image is both the input to the decoder and the target. Masked layers ensure that the conditional distributions for each pixel depends solely on previous ground-truth pixels. (See Appendix G for a recap on autoregressive models). ColTran upsamplers are stacked row/column attention layers that deterministically upsample color and space in parallel. Each attention block (in green) is residual and consists of the following operations: layer-norm → multihead self-attention → MLP.

**Sampling.** The Axial Transformer natively supports semi-parallel sampling that avoids re-evaluation of the entire network to generate each pixel of a RGB image. The encoder is run once per-channel, the outer decoder is run once per-row and the inner decoder is run once per-pixel. The context from the outer decoder and the encoder is initially zero. The encoder conditions the outer decoder (Eq 1) and the encoder + outer decoder condition the inner decoder (Eq 4). The inner decoder then generates a row, one pixel at a time via Eqs. (4) to (6). After generating all pixels in a row, the outer decoder recomputes context via Eqs. (1) to (3) and the inner decoder generates the next row. This proceeds till all the pixels in a channel are generated. The encoder, then recomputes context to generate the next channel.

## 4 PROPOSED ARCHITECTURE

Image colorization is the task of transforming a grayscale image $x^g \in \mathbb{R}^{H \times W \times 1}$ into a colored image $x \in \mathbb{R}^{H \times W \times 3}$. The task is inherently stochastic; for a given grayscale image $x^g$, there exists a conditional distribution over $x$, $p(x|x^g)$. Instead of predicting $x$ directly from $x^g$, we instead sequentially predict two intermediate low resolution images $x^{s\downarrow}$ and $x^{s\downarrow c\downarrow}$ with different color depth first. Besides simplifying the task of high-resolution image colorization into simpler tasks, the smaller resolution allows for training larger models.

We obtain $x^{s\downarrow}$, a spatially downsampled representation of $x$, by standard area interpolation. $x^{s\downarrow c\downarrow}$ is a 3 bit per-channel representation of $x^{s\downarrow}$, that is, each color channel has only 8 intensities. Thus, there are $8^3 = 512$ coarse colors per pixel which are predicted directly as a single "color" channel. We rewrite the conditional likelihood $p(x|x^g)$ to incorporate the intermediate representations as follows:

$$p(x|x^g) = p(x|x^g) \cdot 1 = p(x|x^g) \cdot p(x^{s\downarrow c\downarrow}, x^{s\downarrow}|x, x^g) = p(x^{s\downarrow c\downarrow}, x^{s\downarrow}, x|x^g) \tag{7}$$

$$= p(x|x^{s\downarrow}, x^g) \cdot p(x^{s\downarrow}|x^{s\downarrow c\downarrow}, x^g) \cdot p(x^{s\downarrow c\downarrow}|x^g) \tag{8}$$

ColTran core (Section 4.1), a parallel color upsampler and a parallel spatial upsampler (Section 4.3) model $p(x^{s\downarrow c\downarrow}|x^g), p(x^{s\downarrow}|x^{s\downarrow c\downarrow}, x^g)$ and $p(x|x^{s\downarrow})$ respectively. In the subsections below, we describe

| Component | Unconditional | Conditional |
|---|---|---|
| **Self-Attention** | $\mathbf{y} = \text{Softmax}(\frac{\mathbf{q}\mathbf{k}^{\top}}{\sqrt{D}})\mathbf{v}$ | $\mathbf{y} = \text{Softmax}(\frac{\mathbf{q}_c\mathbf{k}_c^{\top}}{\sqrt{D}})\mathbf{v}_c$ 

 where $\quad \forall \mathbf{z} = \mathbf{k}, \mathbf{q}, \mathbf{v}$ 
 $\mathbf{z}_c = (\mathbf{c}U_s^z) \odot \mathbf{z} + (\mathbf{c}U_b^z)$ |
| **MLP** | $\mathbf{y} = \text{ReLU}(\mathbf{x}U_1 + \mathbf{b}_1)U_2 + \mathbf{b}_2$ | $\mathbf{h} = \text{ReLU}(\mathbf{x}U_1 + \mathbf{b}_1)U_2 + \mathbf{b}_2$ 
 $\mathbf{y} = (\mathbf{c}U_s^f) \odot \mathbf{h} + (\mathbf{c}U_b^f)$ |
| **Layer Norm** | $\mathbf{y} = \beta\,\text{Norm}(\mathbf{x}) + \gamma$ | $\mathbf{y} = \beta_c\,\text{Norm}(\mathbf{x}) + \gamma_c$ 

 where $\quad \forall \mu = \beta_c, \gamma_c$ 
 $\mathbf{c} \in \mathbb{R}^{H \times W \times D} \to \hat{\mathbf{c}} \in \mathbb{R}^{HW \times D}$ 
 $\mu = (\mathbf{u} \cdot \hat{\mathbf{c}})U_d^{\mu} \quad \mathbf{u} \in \mathbb{R}^{HW}$ |

**Table 1:** We contrast the different components of unconditional self-attention with self-attention conditioned on context $\mathbf{c} \in \mathbb{R}^{M \times N \times D}$. Learnable parameters specific to conditioning are denoted by $\mathbf{u}$ and $U_. \in \mathbb{R}^{D \times D}$.

these individual components in detail. From now on we will refer to all low resolutions as $M \times N$ and high resolution as $H \times W$. An illustration of the overall architecture is shown in Figure 2.

## 4.1 COLTRAN CORE

In this section, we describe ColTran core, a conditional variant of the Axial Transformer (Ho et al., 2019b) for low resolution coarse colorization. ColTran Core models a distribution $p_c(x^{s\downarrow c\downarrow}|x^g)$ over 512 coarse colors for every pixel, conditioned on a low resolution grayscale image in addition to the colors from previously predicted pixels as per raster order (Eq. 9).

$$p_c(x^{s\downarrow c\downarrow}|x^g) = \prod_{i=1}^{M}\prod_{j=1}^{N} p_c(x_{ij}^{s\downarrow c\downarrow}|x^g, x_{<i}^{s\downarrow c\downarrow}, x_{i,<j}^{s\downarrow c\downarrow}) \tag{9}$$

Given a context representation $\mathbf{c} \in \mathbb{R}^{M \times N \times D}$ we propose conditional transformer layers in Table 1. Conditional transformer layers have conditional versions of all components within the standard attention block (see Appendix H, Eqs. 14-18).

**Conditional Self-Attention.** For every layer in the decoder, we apply six $1 \times 1$ convolutions to $\mathbf{c}$ to obtain three scale and shift vectors which we apply element-wise to $\mathbf{q}$, $\mathbf{k}$ and $\mathbf{v}$ of the self-attention operation (Appendix 3.1), respectively.

**Conditional MLP.** A standard component of the transformer architecture is a two layer pointwise feed-forward network after the self-attention layer. We scale and shift to the output of each MLP conditioned on $\mathbf{c}$ as for self-attention.

**Conditional Layer Norm.** Layer normalization (Ba et al., 2016) globally scales and shifts a given normalized input using learnable vectors $\beta, \gamma$. Instead, we predict $\beta_c$ and $\gamma_c$ as a function of $\mathbf{c}$. We first aggregate $\mathbf{c}$ into a global 1-D representation $\overline{\mathbf{c}} \in \mathbb{R}^L$ via a learnable, spatial pooling layer. Spatial pooling is initialized as a mean pooling layer. Similar to 1-D conditional normalization layers (Perez et al., 2017; De Vries et al., 2017; Dumoulin et al., 2016; Huang & Belongie, 2017), we then apply a linear projection on $\overline{\mathbf{c}}$ to predict $\beta_c$ and $\gamma_c$, respectively.

A grayscale encoder consisting of multiple, alternating row and column self-attention layers encodes the grayscale image into the initial conditioning context $\mathbf{c}^g$. It serves as both context for the conditional layers and as additional input to the embeddings of the outer decoder. The sum of the outer decoder's output and $\mathbf{c}^g$ condition the inner decoder. Figure 2 illustrates how conditioning is applied in the autoregressive core of the ColTran architecture.

Conditioning every layer via multiple components allows stronger gradient signals through the encoder and as an effect the encoder can learn better contextual representations. We validate this empirically by outperforming the native Axial Transformer that conditions context states by biasing (See Section 5.2 and Section 5.4).

## 4.2 AUXILIARY PARALLEL MODEL

We additionally found it beneficial to train an auxiliary parallel prediction model that models $\widetilde{p}_c(x^{s\downarrow c\downarrow})$ directly on top of representations learned by the grayscale encoder which we found beneficial for regularization (Eq. 10)

$$\widetilde{p}_c(x^{s\downarrow c\downarrow}|x^g) = \prod_{i=1}^{M}\prod_{j=1}^{N}\widetilde{p}_c(x_{ij}^{s\downarrow c\downarrow}|x^g) \tag{10}$$

Intuitively, this forces the model to compute richer representations and global color structure already at the output of the encoder which can help conditioning and therefore has a beneficial, regularizing effect on learning. We apply a linear projection, $U_{\text{parallel}} \in \mathbb{R}^{L \times 512}$ on top of $\mathbf{c}^g$ (the output of the grayscale encoder) into a per-pixel distribution over 512 coarse colors. It was crucial to tune the relative contribution of the autoregressive and parallel predictions to improve performance which we study in Section 5.3

## 4.3 COLOR & SPATIAL UPSAMPLING

In order to produce high-fidelity colorized images from low resolution, coarse color images and a given high resolution grayscale image, we train color and spatial upsampling models. They share the same architecture while differing in their respective inputs and resolution at which they operate. Similar to the grayscale encoder, the upsamplers comprise of multiple alternating layers of row and column self-attention. The output of the encoder is projected to compute the logits underlying the per pixel color probabilities of the respective upsampler. Figure 2 illustrates the architectures

**Color Upsampler.** We convert the coarse image $x^{s\downarrow c\downarrow} \in \mathbb{R}^{M \times N \times 1}$ of 512 colors back into a 3 bit RGB image with 8 symbols per channel. The channels are embedded using separate embedding matrices to $\mathbf{x}_k^{s\downarrow c\downarrow} \in \mathbb{R}^{M \times N \times D}$, where $k \in \{R, G, B\}$ indicates the channel. We upsample each channel individually conditioning only on the respective channel's embedding. The channel embedding is summed with the respective grayscale embedding for each pixel and serve as input to the subsequent self-attention layers (encoder). The output of the encoder is further projected to per pixel-channel probability distributions $\widetilde{p}_{c\uparrow}(x_k^{s\downarrow}|x^{s\downarrow c\downarrow}, x^g) \in \mathbb{R}^{M \times N \times 256}$ over 256 color intensities for all $k \in \{R, G, B\}$ (Eq. 11).

$$\widetilde{p}_{c\uparrow}(x^{s\downarrow}|x^g) = \prod_{i=1}^{M}\prod_{j=1}^{N}\widetilde{p}_{c\uparrow}(x_{ij}^{s\downarrow}|x^g, x^{s\downarrow c\downarrow}) \tag{11}$$

**Spatial Upsampler.** We first naively upsample $x^{s\downarrow} \in \mathbb{R}^{M \times N \times 3}$ into a blurry, high-resolution RGB image using area interpolation. As above, we then embed each channel of the blurry RGB image and run a per-channel encoder exactly the same way as with the color upsampler. The output of the encoder is finally projected to per pixel-channel probability distributions $\widetilde{p}_{s\uparrow}(x_k|x^{s\downarrow}, x^g) \in \mathbb{R}^{H \times W \times 256}$ over 256 color intensities for all $k \in \{R, G, B\}$. (Eq. 12)

$$\widetilde{p}_{s\uparrow}(x|x^g) = \prod_{i=1}^{H}\prod_{j=1}^{W}\widetilde{p}_{s\uparrow}(x_{ij}|x^g, x^{s\downarrow}) \tag{12}$$

In our experiments, similar to (Guadarrama et al., 2017), we found parallel upsampling to be sufficient for high quality colorizations. Parallel upsampling has the huge advantage of fast generation which would be notoriously slow for full autoregressive models on high resolution. To avoid plausible minor color inconsistencies between pixels, instead of sampling each pixel from the predicted distribution in (Eq. 12 and Eq. 11), we just use the argmax. Even though this slightly limits the potential diversity of colorizations, in practice we observe that sampling only coarse colors via ColTran core is enough to produce a great variety of colorizations.

**Objective.** We train our architecture to minimize the negative log-likelihood (Eq. 13) of the data. $p_c/\widetilde{p}_c$, $\widetilde{p}_{s\uparrow}$, $\widetilde{p}_{c\uparrow}$ are maximized independently and $\lambda$ is a hyperparameter that controls the relative contribution of $p_c$ and $\widetilde{p}_c$

$$\mathcal{L} = (1-\lambda)\log p_c + \lambda \log \widetilde{p}_c + \log \widetilde{p}_{c\uparrow} + \log \widetilde{p}_{s\uparrow} \tag{13}$$

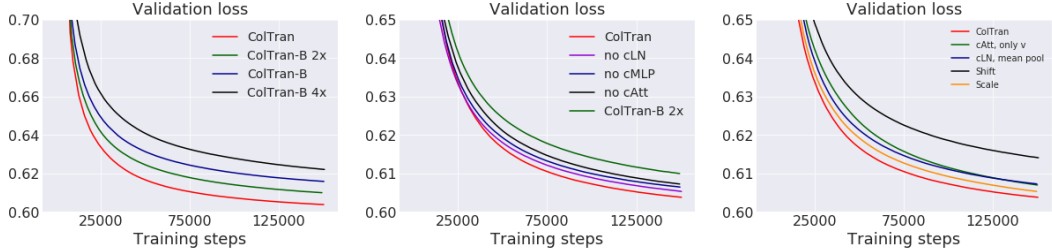

**Figure 3:** Per pixel log-likelihood of coarse colored $64 \times 64$ images over the validation set as a function of training steps. We ablate the various components of the ColTran core in each plot. **Left:** *ColTran* with Conditional Transformer Layers vs a baseline Axial Transformer which conditions via addition (*ColTran-B*). *ColTran-B 2x* and *ColTran-B 4x* refer to wider baselines with increased model capacity. **Center:** Removing each conditional sub-component one at a time (*no cLN*, *no cMLP* and *no cAtt*). **Right:** Conditional shifts only (*Shift*), Conditional scales only (*Scale*), removal of kq conditioning in cAtt (*cAtt, only v*) and fixed mean pooling in cLN (*cLN, mean pool*). See Section 5.2 for more details.

## 5 EXPERIMENTS

### 5.1 TRAINING AND EVALUATION

We evaluate ColTran on colorizing $256 \times 256$ grayscale images from the ImageNet dataset (Russakovsky et al., 2015). We train the ColTran core, color and spatial upsamplers independently on 16 TPUv2 chips with a batch-size of 224, 768 and 32 for 600K, 450K and 300K steps respectively. We use 4 axial attention blocks in each component of our architecture, with a hidden size of 512 and 4 heads. We use RMSprop (Tieleman & Hinton, 2012) with a fixed learning rate of $3e - 4$. We set apart 10000 images from the training set as a holdout set to tune hyperparameters and perform ablations. To compute FID, we generate 5000 samples conditioned on the grayscale images from this holdout set. We use the public validation set to display qualitative results and report final numbers.

### 5.2 ABLATIONS OF COLTRAN CORE

The autoregressive core of ColTran models downsampled, coarse-colored images of resolution $64 \times 64$ with 512 coarse colots, conditioned on the respective grayscale image. In a series of experiments we ablate the different components of the architecture (Figure 3). In the section below, we refer to the conditional self-attention, conditional layer norm and conditional MLP subcomponents as cAtt, cLN and cMLP respectively. We report the per-pixel log-likelihood over 512 coarse colors on the validation set as a function of training steps.

**Impact of conditional transformer layers.** The left side of Figure 3 illustrates the significant improvement in loss that ColTran core (with conditional transformer layers) achieves over the original Axial Transformer (marked *ColTran-B*). This demonstrates the usefulness of our proposed conditional layers. Because conditional layers introduce a higher number of parameters we additionally compare to and outperform the original Axial Transformer baselines with 2x and 4x wider MLP dimensions (labeled as *ColTran-B 2x* and *ColTran-B 4x*). Both *ColTran-B 2x* and *ColTran-B 4x* have an increased parameter count which makes for a fair comparison. Our results show that the increased performance cannot be explained solely by the fact that our model has more parameters.

**Importance of each conditional component.** We perform a leave-one-out study to determine the importance of each conditional component. We remove each conditional component one at a time and retrain the new ablated model. The curves *no cLN*, *no cMLP* and *no cAtt* in the middle of Figure 3 quantifies our results. While each conditional component improves final performance, cAtt plays the most important role.

**Multiplicative vs Additive Interactions.** Conditional transformer layers employ both conditional shifts and scales consisting of additive and multiplicative interactions, respectively. The curves *Scale* and *Shift* on the right hand side of Figure 3 demonstrate the impact of these interactions via ablated architectures that use conditional shifts and conditional scales only. While both types of interactions are important, multiplicative interactions have a much stronger impact.

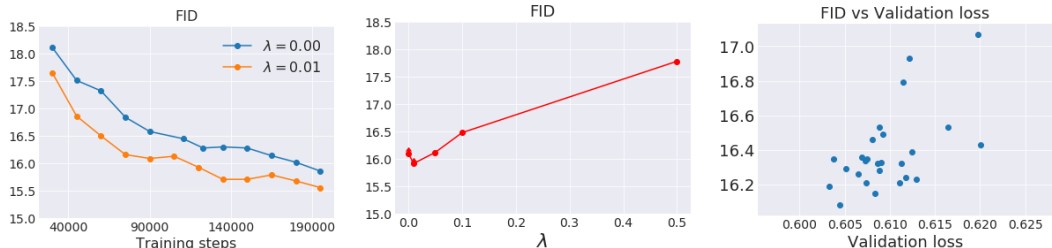

**Figure 4: Left:** FID of generated $64 \times 64$ coarse samples as a function of training steps for $\lambda = 0.01$ and $\lambda = 0.0$. **Center:** Final FID scores as a function of $\lambda$. **Right:** FID as a function of log-likelihood.

**Context-aware dot product attention.** Self-attention computes the similarity between pixel representations using a dot product between $\mathbf{q}$ and $\mathbf{k}$ (See: Eq 15). cAtt applies conditional shifts and scales on $\mathbf{q}$, $\mathbf{k}$ and allow modifying this similarity based on contextual information. The curve *cAtt, only v* on the right of Figure 3 shows that removing this property, by conditioning only on $\mathbf{v}$ leads to worse results.

**Fixed vs adaptive global representation:** cLN aggregates global information with a flexible learnable spatial pooling layer. We experimented with a fixed mean pooling layer forcing all the cLN layers to use the same global representation with the same per-pixel weight. The curve *cLN, mean pool* on the right of Figure 3 shows that enforcing this constraint causes inferior performance as compared to even having no cLN. This indicates that different aggregations of global representations are important for different cLN layers.

## 5.3 OTHER ABLATIONS

**Auxiliary Parallel Model.** We study the effect of the hyperparameter $\lambda$, which controls the contribution of the auxiliary parallel prediction model described in Section 4.2. For a given $\lambda$, we now optimize $\hat{p}_c(\lambda) = (1 - \lambda) \log p_c(.) + \lambda \log \widetilde{p}_c(.)$ instead of just $\log p_c(.)$. Note that $\widetilde{p}_c(.)$, models each pixel independently, which is more difficult than modelling each pixel conditioned on previous pixels given by $p_c(.)$. Hence, employing $\hat{p}_c(\lambda)$ as a holdout metric, would just lead to a trivial soluion at $\lambda = 0$. Instead, the FID of the generated coarse 64x64 samples provides a reliable way to find an optimal value of $\lambda$. In Figure 4, at $\lambda = 0.01$, our model converges to a better FID faster with a marginal but consistent final improvement. At higher values the performance deteriorates quickly.

**Upsamplers.** Upsampling coarse colored, low-resolution images to a higher resolution is much simpler. Given ground truth $64 \times 64$ coarse images, the ColTran upsamplers map these to fine grained $256 \times 256$ images without any visible artifacts and FID of 16.4. For comparison, the FID between two random sets of 5000 samples from our holdout set is 15.5. It is further extremely important to provide the grayscale image as input to each of the individual upsamplers, without which the generated images appear highly smoothed out and the FID drops to 27.0. We also trained a single upsampler for both color and resolution. The FID in this case drops marginally to 16.6.

## 5.4 FRECHET INCEPTION DISTANCE

We compute FID using colorizations of 5000 grayscale images of resolution $256 \times 256$ from the ImageNet validation set as done in (Ardizzone et al., 2019). To compute the FID, we ensure that there is no overlap between the grayscale images that condition ColTran and those in the ground-truth distribution. In addition to ColTran, we report two additional results *ColTran-S* and *ColTran-B*. *ColTran-B* refers to the baseline Axial Transformer that conditions via addition at the input. PixColor samples smaller $28 \times 28$ colored images autoregressively as compared to ColTran's $64 \times 64$. As a control experiment, we train an autoregressive model on resolution $28 \times 28$ (*ColTran-S*) to disentangle architectural choices and the inherent stochasticity of modelling higher resolution images. *ColTran-S* and *ColTran-B* obtains FID scores of 22.06 and 19.98 that significantly improve over the previous best FID of 24.32. Finally, ColTran achieves the best FID score of 19.37. All results are presented in Table 2 left.

| Models | FID |
|---|---|
| ColTran | **19.37 ± 0.09** |
| ColTran-B | 19.98 ± 0.20 |
| ColTran-S | 22.06 ± 0.13 |
| PixColor [16] | 24.32 ± 0.21 |
| cGAN [3] | 24.41 ± 0.27 |
| cINN [1] | 25.13 ± 0.3 |
| VAE-MDN [11] | 25.98 ± 0.28 |
| Ground truth | 14.68 ± 0.15 |
| Grayscale | 30.19 ± 0.1 |

| Models | AMT Fooling rate |
|---|---|
| ColTran (Oracle) | 62.0 % ± 0.99 |
| ColTran (Seed 1) | 40.5 % ± 0.81 |
| ColTran (Seed 2) | **42.3 % ± 0.76** |
| ColTran (Seed 3) | 41.7 % ± 0.83 |
| PixColor [16] (Oracle) | 38.3 % ± 0.98 |
| PixColor (Seed 1) | 33.3 % ± 1.04 |
| PixColor (Seed 2) | 35.4 % ± 1.01 |
| PixColor (Seed 3) | 33.2 % ± 1.03 |
| CIC [56] | 29.2 % ± 0.98 |
| LRAC [27] | 30.9 % ± 1.02 |
| LTBC [22] | 25.8 % ± 0.97 |

**Table 2:** We outperform various state-of-the-art colorization models both on FID (left) and human evaluation (right). We obtain the FID scores from (Ardizzone et al., 2019) and the human evaluation results from (Guadarrama et al., 2017). ColTran-B is a baseline Axial Transformer that conditions via addition and ColTran-S is a control experiment where we train ColTran core (See: 4.1) on smaller $28 \times 28$ colored images.

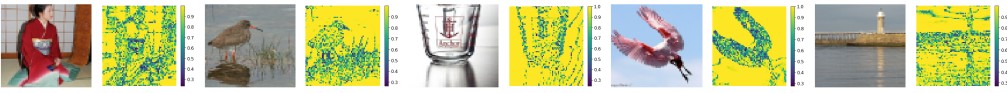

**Figure 5:** We display the per-pixel, maximum predicted probability over 512 colors as a proxy for uncertainty.

**Correlation between FID and Log-likelihood.** For each architectural variant, Figure 4 right illustrates the correlation between the log-likelihood and FID after 150K training steps. There is a moderately positive correlation of 0.57 between the log-likelihood and FID. Importantly, even an absolute improvement on the order of 0.01 - 0.02 can improve FID significantly. This suggests that designing architectures that achieve better log-likelihood values is likely to lead to improved FID scores and colorization fidelity.

## 5.5 QUALITATIVE EVALUATION

**Human Evaluation.** For our qualitative assessment, we follow the protocol used in PixColor (Guadarrama et al., 2017). ColTran colorizes 500 grayscale images, with 3 different colorizations per image, denoted as seeds. Human raters assess the quality of these colorizations with a two alternative-forced choice (2AFC) test. We display both the ground-truth and recolorized image sequentially for one second in random order. The raters are then asked to identify the image with fake colors. For each seed, we report the mean fooling rate over 500 colorizations and 5 different raters. For the oracle methods, we use the human rating to pick the best-of-three colorizations. ColTran's best seed achieves a fooling rate of 42.3 % compared to the 35.4 % of PixColor's best seed. ColTran Oracle achieves a fooling rate of 62 %, indicating that human raters prefer ColTran's best-of-three colorizations over the ground truth image itself.

**Visualizing uncertainty.** The autoregressive core model of ColTran should be highly uncertain at object boundaries when colors change. Figure 5 illustrates the per-pixel, maximum predicted probability over 512 colors as a proxy for uncertainty. We observe that the model is indeed highly uncertain at edges and within more complicated textures.

## 6 CONCLUSION

We presented the Colorization Transformer (ColTran), an architecture that entirely relies on self-attention for image colorization. We introduce conditional transformer layers, a novel building block for conditional, generative models based on self-attention. Our ablations show the superiority of employing this mechanism over a number of different baselines. Finally, we demonstrate that ColTran can generate diverse, high-fidelity colorizations on ImageNet, which are largely indistinguishable from the ground-truth even for human raters.

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

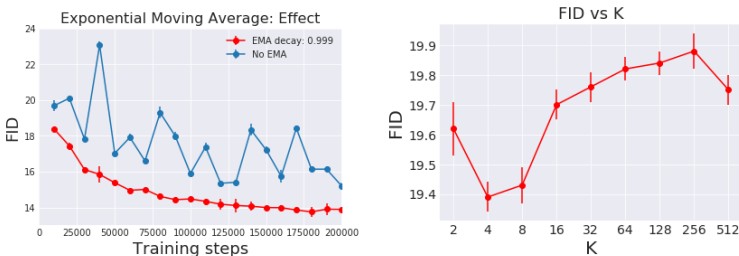

**Figure 6: Left**: FID vs training steps, with and without polyak averaging. **Right**: The effect of K in top-K sampling on FID. See Appendix B and E

ACKNOWLEDGEMENTS

We would like to thank Mohammad Norouzi, Rianne van den Berg, Mostafa Dehghani for their useful comments on the draft and Avital Oliver for assistance in the Mechanical Turk setup.

CHANGELOG

- **v2:** Dataset Sharding fix across multiple TPU workers. This changed the FID scores of ColTran, ColTran-B and ColTran-S from their **v1** values of 19.71, 21.6 and 21.9 to their **v2** values of 19.37, 19.98 and 22.06 respecitvely.

## A  CODE, CHECKPOINTS AND TENSORBOARD FILES

Our implementation is open-sourced in the google-research framework at https://github.com/google-research/google-research/tree/master/coltran with a zip compressed version here. Our full set of hyperparameters are available here.

We provide pre-trained checkpoints of the colorizer and upsamplers on ImageNet at https://console.cloud.google.com/storage/browser/gresearch/coltran. Finally, reference tensorboard files for our training runs are available at colorizer tensorboard, color upsampler tensorboard and spatial upsampler tensorboard.

## B  EXPONENTIAL MOVING AVERAGE

We found using an exponential moving average (EMA) of our checkpoints, extremely crucial to generate high quality samples. In Figure 6, we display the FID as a function of training steps, with and without EMA. On applying EMA, our FID score improves steadily over time.

## C  NUMBER OF PARAMETERS AND INFERENCE SPEED

**Inference speed.**   ColTran core can sample a batch of 20 64x64 grayscale images in around 3.5 -5 minutes on a P100 GPU vs PixColor that takes 10 minutes to colorize 28x28 grayscale images on a K40 GPU. Sampling 28x28 colorizations takes around 30 seconds. The upsampler networks take in the order of milliseconds.

Further, in our naive implementation, we recompute the activations, $\mathbf{c}U_s^z, \mathbf{c}U_b^z, \mathbf{c}U_s^f, \mathbf{c}U_b^f$ in Table 1 to generate every pixel in the inner decoder. Instead, we can compute these activations once per-grayscale image in the encoder and once per-row in the outer decoder and reuse them. This is likely to speed up sampling even more and we leave this engineering optimization for future work.

**Number of parameters.**   ColTran has a total of ColTran core (46M) + Color Upsampler (14M) + Spatial Upsampler (14M) = 74M parameters. In comparison, PixColor has Conditioning network (44M) + Colorizer network (11M) + Refinement Network (28M) = 83M parameters.

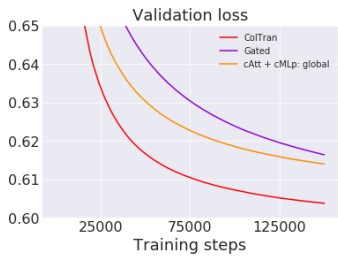

**Figure 7:** Ablated models. *Gated*: Gated conditioning layers as done in (Oord et al., 2016) and *cAtt + cMLP, global*: Global conditioning instead of pointwise conditioning in cAtt and cLN.

## D   LOWER COMPUTE REGIME

We retrained the autoregressive colorizer and color upsampler on 4 TPUv2 chips (the lowest configuration) with a reduced-batch size of 56 and 192 each. For the spatial upsampler, we found that a batch-size of 8 was sub-optimal and lead to a large deterioration in loss. We thus used a smaller spatial upsampler with 2 axial attention blocks with a batch-size of 16 and trained it also on 4 TPUv2 chips. The FID drops from 19.71 to 20.9 which is still significantly better than the other models in 2. We note that in this experiment, we use only 12 TPUv2 chips in total while PixColor (Guadarrama et al., 2017) uses a total of 16 GPUs.

## E   IMPROVED FID WITH TOP-K SAMPLING

We can improve colorization fidelity and remove artifacts due to unnatural colors via Top-K sampling at the cost of reduced colorization diversity. In this setting, for a given pixel ColTran generates a color from the top-K colors (instead of 512 colors) as determined by the predicted probabilities. Our results in Figure 6 $K = 4$ and $K = 8$ demonstrate a performance improvement over the baseline ColTran model with $K = 512$

## F   ADDITIONAL ABLATIONS:

Additional ablations of our conditional transformer layers are in Figure 7 which did not help.

- Conditional transformer layers based on Gated layers (Oord et al., 2016) (*Gated*)
- A global conditioning layer instead of pointwise conditioning in cAtt and cLN. *cAtt + cMLP, global*

.

## G   AUTOREGRESSIVE MODELS

Autoregressive models are a family of probabilistic methods that model joint distribution of data $P(x)$ or a sequence of symbols $(x_1, x_2, \ldots x_n)$ as a product of conditionals $\prod_{i=1}^{N} P(x_i|x_{<i})$. During training, the input to autoregressive models are the entire sequence of ground-truth symbols. Masking ensures that the contribution of all "future" symbols in the sequence are zeroed out. The outputs of the autoregressive model are the corresponding conditional distributions. $P(x_i|x_{<i})$. Optimizing the parameters of the autoregressive model proceeds by a standard log-likelihood objective.

Generation happens sequentially, symbol-by-symbol. Once a symbol $x_i$ is generated, the entire sequence $(x_1, x_2, \ldots x_i)$ are fed to the autoregressive model to generate $x_{i+1}$.

In the case of autoregressive image generation symbols typically correspond to the 3 RGB pixel-channel. These are generated sequentially in raster-scan order, channel by channel and pixel by pixel.

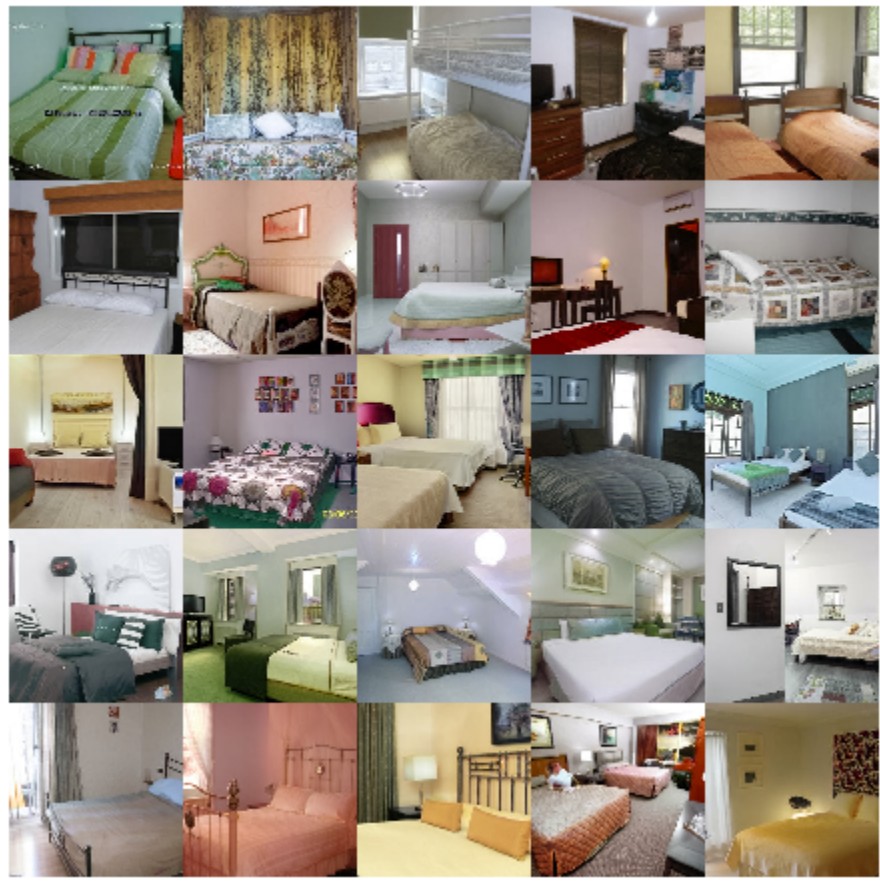

**Figure 8:** We train our colorization model on ImageNet and display high resolution colorizations from LSUN

## H    ROW/COLUMN SELF-ATTENTION

In the following we describe row self-attention, that is, we omit the height dimension as all operations are performed in parallel for each column. Given the representation of a single row within of an image $\mathbf{x}_{i,\cdot} \in \mathbb{R}^{W \times D}$, row-wise self-attention block is applied as follows:

$$[\mathbf{q}, \mathbf{k}, \mathbf{v}] = \mathrm{LN}(\mathbf{x}_{i,\cdot})U_{qkv} \qquad\qquad U_{qkv} \in \mathbb{R}^{D \times 3D_h} \qquad (14)$$

$$A = \mathrm{softmax}\left(\mathbf{q}\mathbf{k}^\top / \sqrt{D_h}\right) \qquad\qquad A \in \mathbb{R}^{W \times W} \qquad (15)$$

$$\mathrm{SA}(\mathbf{x}_{i,\cdot}) = A\mathbf{v} \qquad (16)$$

$$\mathrm{MSA}(\mathbf{x}_{i,\cdot}) = [\mathrm{SA}_1(\mathbf{x}_{i,\cdot}), \mathrm{SA}_2(\mathbf{x}_{i,\cdot}), \cdots, \mathrm{SA}_k(\mathbf{x}_{i,\cdot})]\, U_{out} \qquad U_{out} \in \mathbb{R}^{k \cdot D_h \times D} \qquad (17)$$

LN refers to the application of layer normalization (Ba et al., 2016). Finally, we apply residual connections and a feed-forward neural network with a single hidden layer and ReLU activation (MLP) after each self-attention block as it is common practice in transformers.

$$\hat{\mathbf{x}}_{i,\cdot} = \mathrm{MLP}(\mathrm{LN}(\mathbf{x}'_{i,\cdot})) + \mathbf{x}'_{i,\cdot} \qquad\qquad \mathbf{x}'_{i,\cdot} = \mathrm{MSA}(\mathbf{x}_{i,\cdot}) + \mathbf{x}_{i,\cdot} \qquad (18)$$

Column-wise self-attention over $\mathbf{x}_{\cdot,j} \in \mathbb{R}^{H \times D}$ works analogously.

## I    OUT OF DOMAIN COLORIZATIONS

We use our trained colorization model on ImageNet to colorize high-resolution grayscale images from LSUN $256 \times 256$ (Yu et al., 2015) and low-resolution grayscale images from Celeb-A (Liu

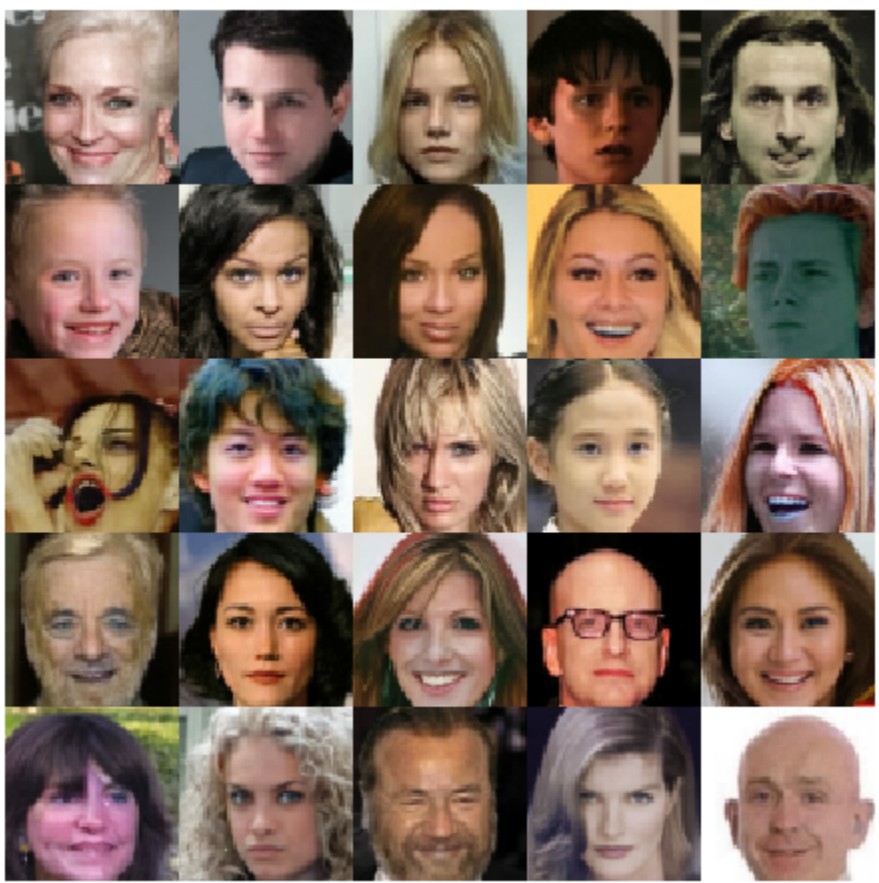

**Figure 9:** We train our colorization model on ImageNet and display low resolution colorizations from Celeb-A

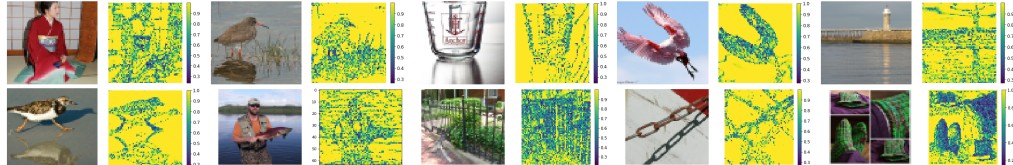

**Figure 10: Top**: Colorizations **Bottom**: Ground truth. From left to right, our colorizations have a progressively higher fooling rate.

et al., 2015) $64 \times 64$. Note that these models were trained only on ImageNet and not finetuned on Celeb-A or LSUN.

## J   NUMBER OF AXIAL ATTENTION BLOCKS

We did a very small hyperparameter sweep using the baseline axial transformer (no conditional layers) with the following configurations:

- hidden size = 512, number of blocks = 4
- hidden size = 1024, number of blocks = 2
- hidden size = 512, number of blocks = 2

Once we found the optimal configuration, we fixed this for all future architecture design.

## K   ANALYSIS OF MTURK RATINGS

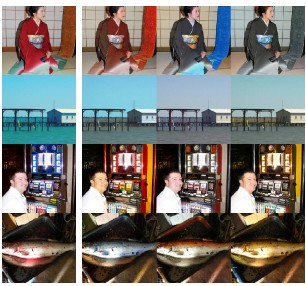 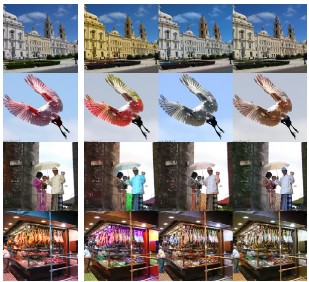 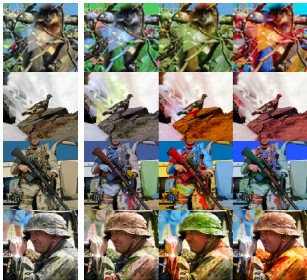

**Figure 11:** In each column, we display the ground truth followed by 3 samples. **Left:** Diverse and real. **Center:** Realism improves from left to right. **Right:** Failure cases

**Figure 12:** We display the per-pixel, maximum predicted probability over 512 colors as a proxy for uncertainty.

We analyzed our samples on the basis of the MTurk ratings in Figure 11. To the left, we show images, where all the samples have a fool rate > 60 %. Our model is able to show diversity in color for both high-level structure and low-level details. In the center, we display samples that have a high variance in MTurk ratings, with a difference of 80 % between the best and the worst sample. All of these are complex objects, that our model is able to colorize reasonably well given multiple attempts. To the right of Figure 11, we show failure cases where all samples have a fool rate of 0 %, For these cases, our model is unable to colorize highly complex structure, that would arguably be difficult even for a human.

## L    MORE PROBABILITY MAPS

We display additional probability maps to visualize uncertainty as done in 5.5.

## M    MORE SAMPLES

We display a wide-diversity of colorizations from ColTran that were not cherry-picked.

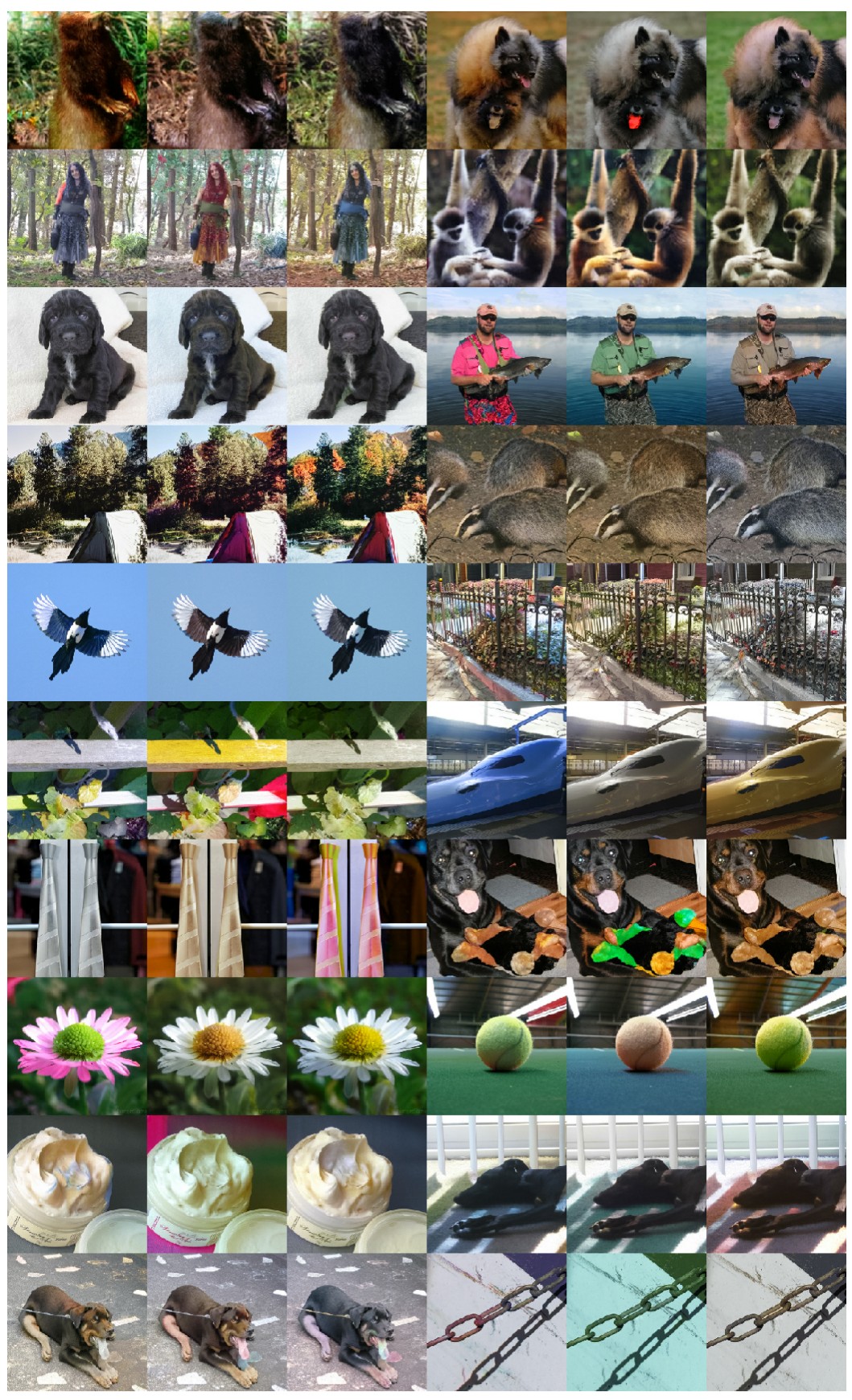

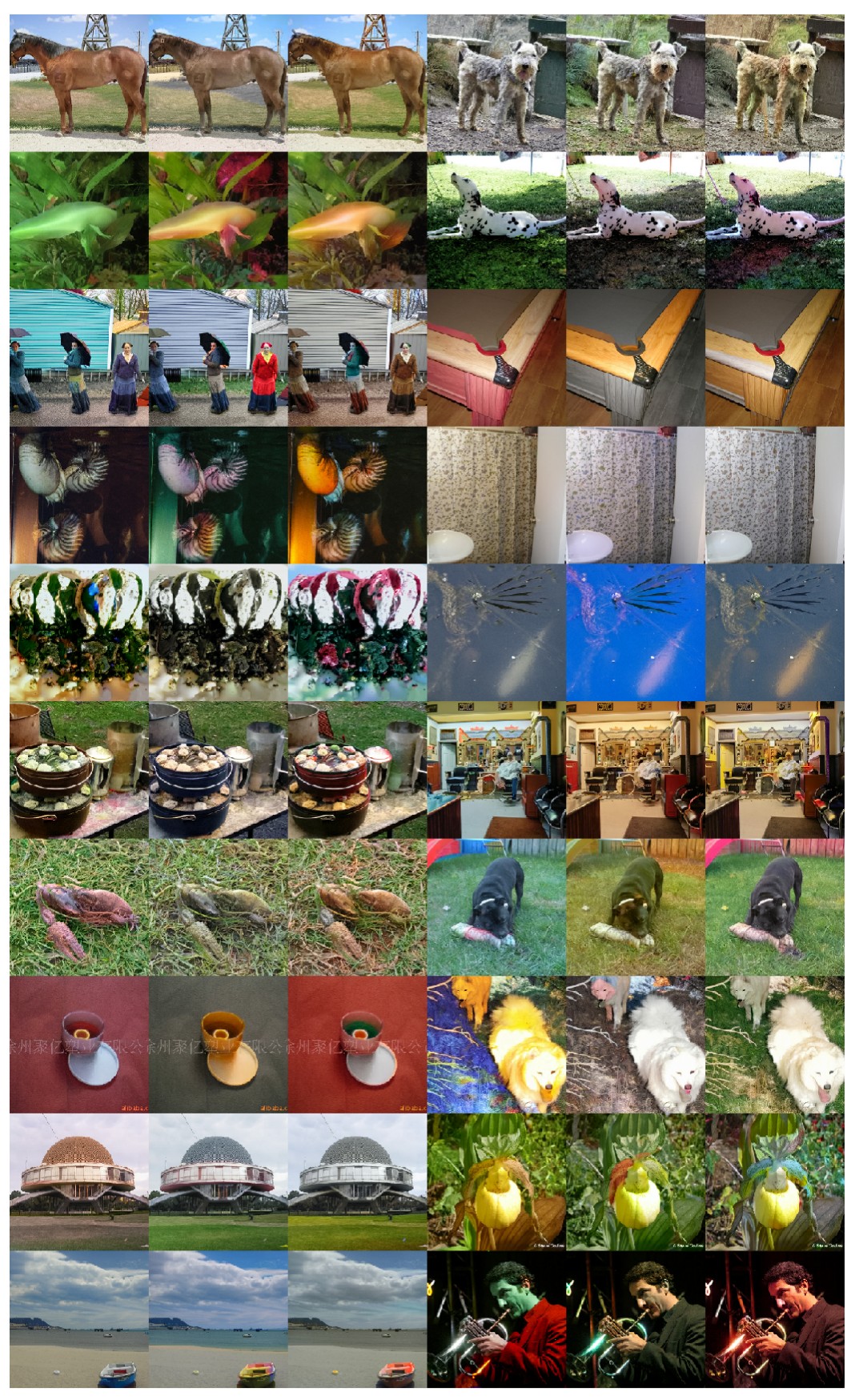

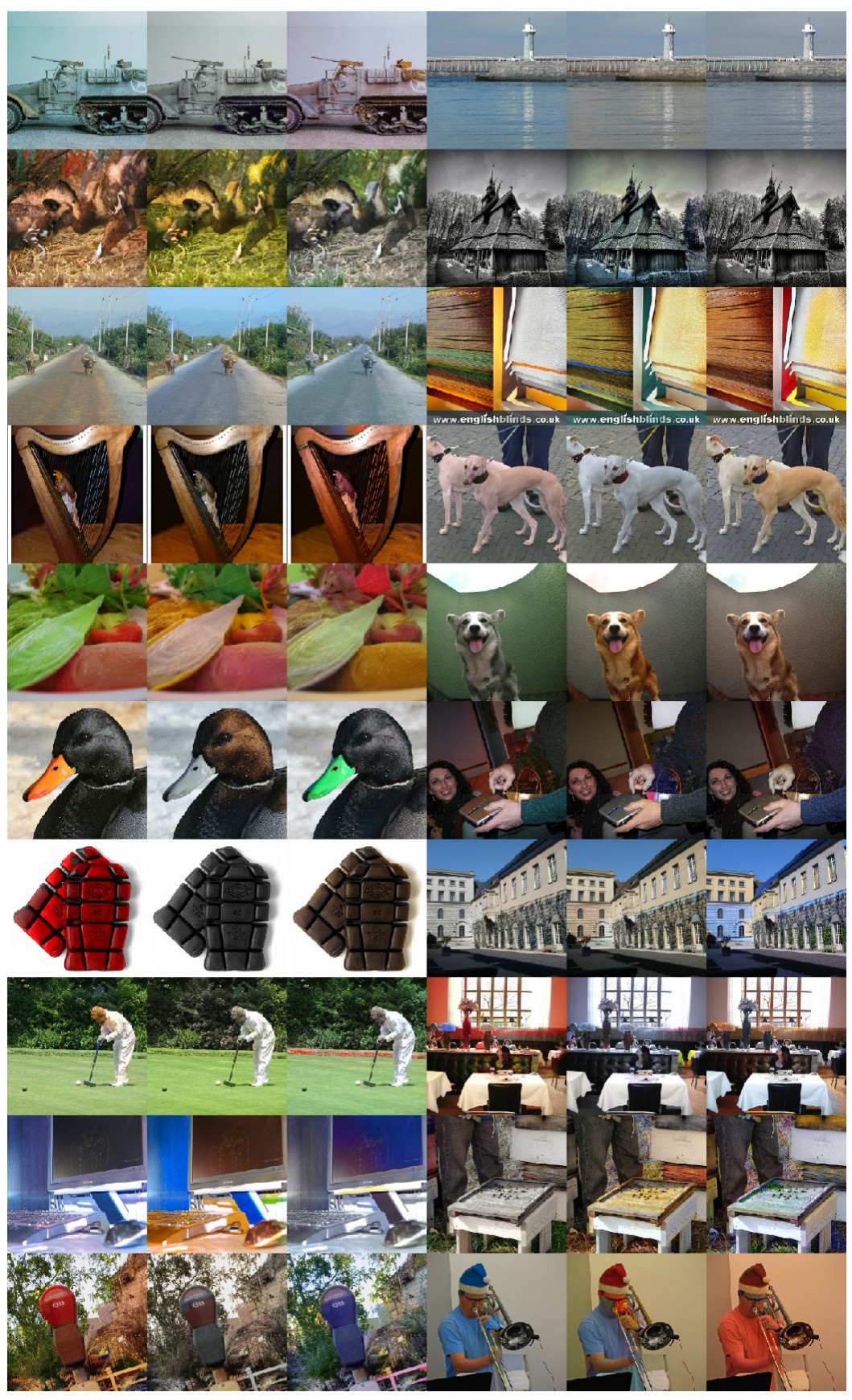

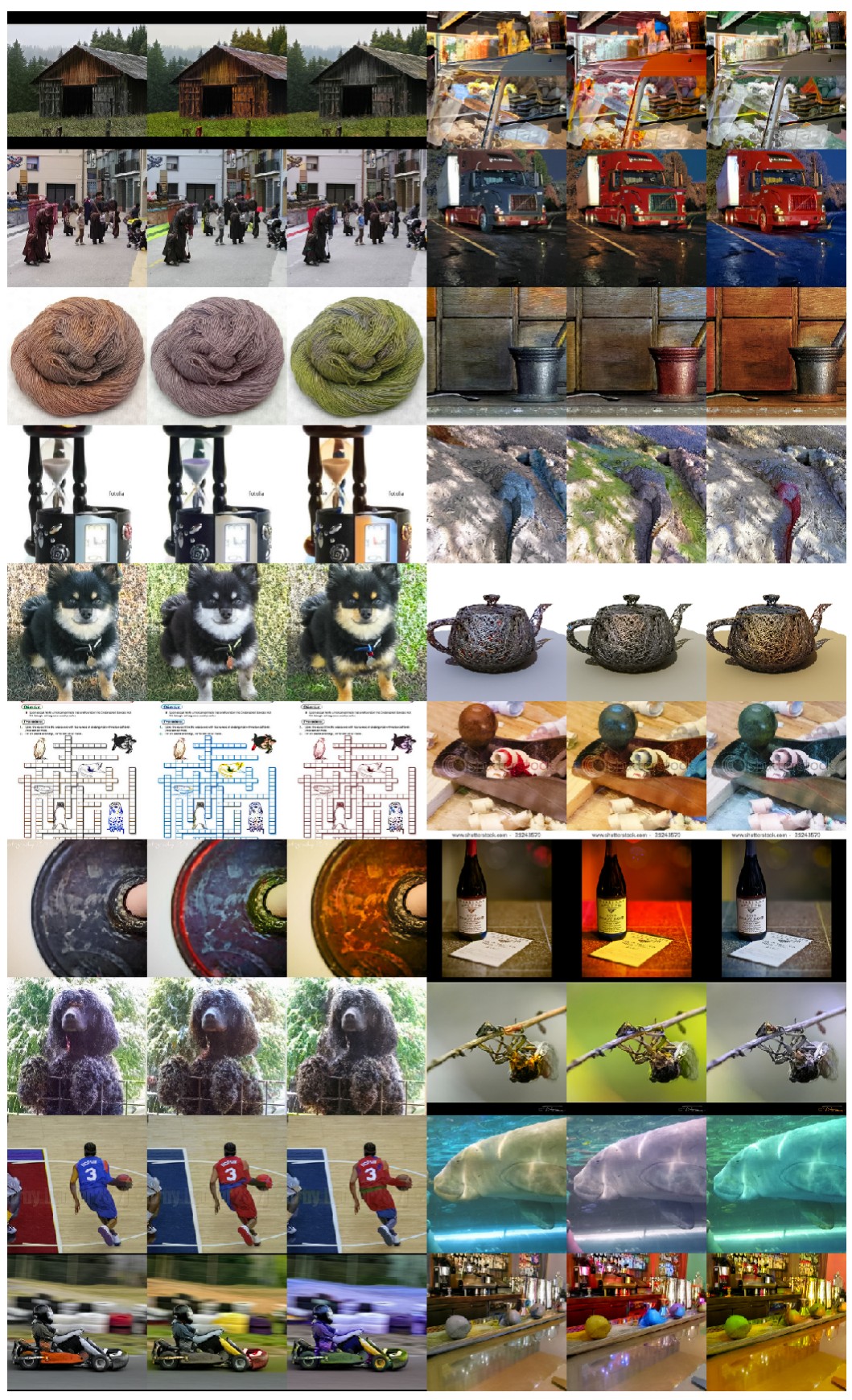

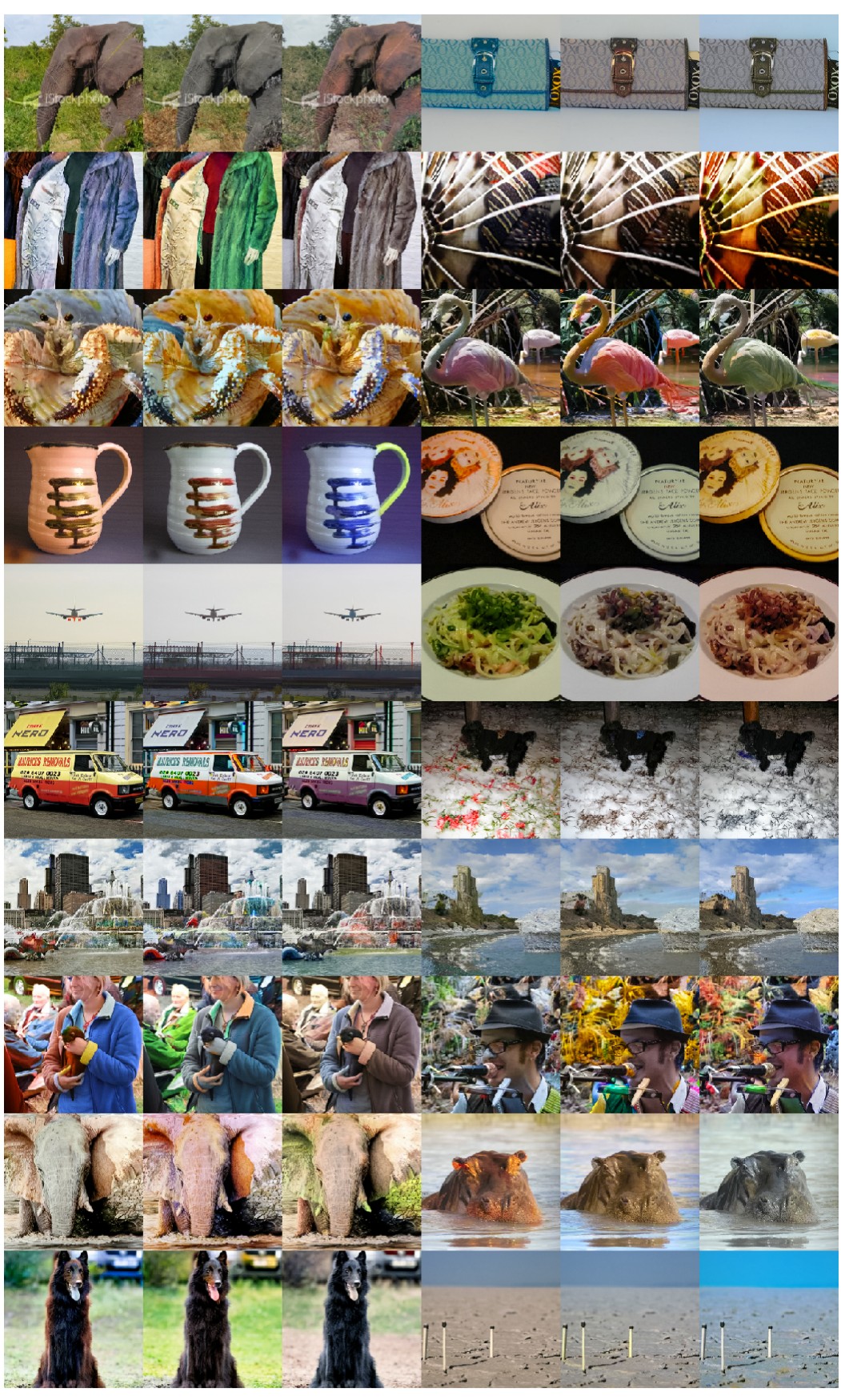

