# OpenReview forum: "Colorization Transformer"
_ICLR.cc/2021/Conference — ICLR 2021 Poster_

### Official Review · AnonReviewer1 · 2020-10-22
**Extensive experiments and strong performance, novelty is a bit incremental.**

**Rating:** 7
**Confidence:** 4

**Review:**

Update: Thanks for the additional ablation studies. I would like to keep my original evaluation which is acceptance.

--------------------

This paper proposes a transformer architecture for image colorization. It uses an axial transformer to process the low-resolution grayscale image, and uses a conditional version of the axial transformer to predict a low-resolution color image autoregressively conditioned on the gray image. It then uses an axial transformer to predict the final high-resolution output pixels.

Pros:
+ The paper is well-written and easy to read. The literature review is comprehensive.
+ Image colorization is an important problem in computer vision. To my knowledge, this is the first paper that applies Transformer to colorization. It could potentially be very impacted and inspire future work.
+ Both automatic metric (FID) and human evaluation are used to compare the method with existing approaches. The performance of the proposed method significantly outperforms the previous state of the art. The qualitative examples are very impressive as well.
+ The paper performs extensive ablation studies (Figure 3) to verify the contribution of different components.

Cons:
- The technical novelty of this paper is a bit limited. It basically applies existing conditioning techniques to the axial transformer and uses it for image colorization.
- It seems that no cLN (Fig. 3 mid) is better than cLN with mean-pool only (Fig. 3 right), which is a bit counterintuitive. Any possible explanation? Also, is there a reason to use the globally aggregated context for cLN but not for cMLP/cAtt? An ablation study on that would be helpful. Besides, there is an ablation study on shift-only modulation but I am curious about how scale-only modulation performs.
- It would be nice to show the number of parameters, training/inference speed of the proposed approach, and compare them to the baselines.
- Please add references to all baseline methods compared in Table 2. I'm able to find the citation of PixColor in other parts of the paper, but cannot find most of the others'.

Minor problems that do not affect my score:
- P1: determinisitic -> deterministic
- The aggregated context is denoted as \hat{c} in Table 1 but as \bar{c} in section 4.3.
- It would be better to use the vector format for Figure 3/4, and enlarge Figure 5 a bit.

Overall, I vote for acceptance. The novelty is not huge but I still think it would be a nice paper for ICLR and have impacts on the field given its strong empirical performance.

---

> ### Author Response · Authors · 2020-11-18
> **Review Response: AnonReviewer 1**
>
> Thanks for your reviews. Please find attached our response.
>
> **Additional ablations**
> * **Scale-only modulation** - We added this curve to **Figure 3**. Scale only modulations perform much better than shift only modulations. Our intuition is that scaling allows to increase or decrease per-pixel activations more easily as compared to biasing. We speculate that this could be useful (for eg to turn off the contributions of individual pixels based on the context when we compute dot products for self-attention)
> * **Global context for cMLP / cATT** - We added this curve to **Appendix G**. Our model performs much worse.
> Our intuition follows from 1). A global context for cAtt means that all key, query and value pairs are scaled/biased by constant values $c_k, c_q$ and $c_v$ The dot-product between k and q would be either $c_k * c_q * k \cdot q$ (for scaling) and $k \cdot q + c_k \sum{q} + c_q \sum{k} + c_k * c_q$. We speculate that this can have a net-effect in increasing the magnitude of dot-product attention and may lead to difficulties in optimization. Elementwise operations (as done in cAtt) are more flexible. For cMLP this effect is purely empirical.
> * **cLN with mean pooling performing worse than no cLN** We expanded a bit on this **Section 5.1**
> A fixed mean pooling layer forces all the cLN layers to use the same global representation with the same weightage per-pixel. The ablation indicates, it is likely that different global representations are meaningful for different cLN layers. Allowing the per-pixel weights to be learnable offers some degrees of freedom and hence different LayerNorm layers can make use of different aggregated global representations.
>
> **Number of parameters / training / inference speed**
> We added a small section in **Appendix H** comparing PixColor to ColTran
> * **Training parameters**:  ColTran has a total of ColTran core (46M) + Color Upsampler (14M) + Spatial Upsampler (14M) = 74M parameters. This is lesser than PixColor that has Conditioning network (44M) + Colorizer network (11M) + Refinement Network (28M) = 83M parameters.
> * **Inference speed** ColTran core can sample 64x64 grayscale images in 4-5 minutes P100 GPU vs PixColor that takes ~10 minutes to colorize 28x28 grayscale images on a K40 GPU. Sampling 28x28 colorizations takes just around 30 seconds. The upsampler networks take in the order of milliseconds.
> * **Training time** Our training time is comparable to PixColor (~3 days). However, we are able to reach FID scores of 20.38 within a single day as compared to PixColor’s final FID 24.32
>
> **References**
> We added references to all baselines in Table 2. They are described in the related work section.
>
> **Technical novelty**
> We added a section to the introduction better highlighting the contributions of this paper.
> * First application of transformers for high-resolution ($256 \times 256$) image colorization.
> * We introduce conditional transformer layers for low-resolution coarse colorization in Section 4.1. The conditional layers incorporate conditioning information via multiple learnable components that are applied per-pixel and per-channel. We validate the contribution of each component with extensive experimentation and ablation studies.
> * We propose training an auxiliary parallel prediction model jointly with the low resolution coarse colorization model in Section 4.2. Improved FID scores demonstrate the usefulness of this auxiliary model.

---

### Official Review · AnonReviewer4 · 2020-10-28
**Image colorization based on self-attention**

**Rating:** 6
**Confidence:** 4

**Review:**

--- Update ---
The authors have addressed several concerns that I had regarding the work.  While this is largely an application of a previous method, they have made some application-specific decisions in order to achieve the significant boost in performance on colorization that they saw.  While the metrics for this task are much improved, there are still some things to be desired on the qualitative results (i.e. the diversity of results tends to be in blocks, as opposed to high within-image color variability).  Nevertheless, I think the improvement from this approach my guide future work in this area.  Given the author's responses and changes made, I have amended my recommendation accordingly.

__1.  Summary__
The authors propose a method for image colorization based on self-attention largely following the architecture of the Axial Transformer (Ho et al., 2019b).  This approach outperforms several SOTA colorization models on FID and human evaluation.


__2a. Strong Points__
The motivation for this work is clear.  Image colorization has many applications and while past approaches have significantly advanced in the past few years, there is certainly much left to be explored in this space.

The recap/explanation of the Axial Transformer is clear and concise.  My concern (see below) is not with the articulation of this section, but more on the reliance of an approach that hasn’t been accepted via peer review.

The performance of this method using both FID as well as using human evaluators is compelling.

Breaking the problem of colorization into two intermediate low resolution images is a nice approach for enabling larger models.  One question would be how well a single model would perform if smaller images were all that was required.

The ablation studies show how different components impact the performance.


__2b. Weak Points__
All three modules of this approach are based on method of (Ho et al., 2019b), which is available on arxiv, but was rejected from ICLR 2020.  The current work is focused on the application of that method. This makes for a bit of a tricky situation.  The description of the Axial Transformer is given in section 4, but it is only textual and refers the readers back to the pre-print for more detail.  Since this is the central method of the current work, at a minimum I think it requires more explanation/justification as opposed to pointing to a work that has not been accepted via peer review.

While the language of the paper is fine, the overall flow of the paper is lacking a bit of narrative.  Overall I found myself having to jump around to find the definition and explanation of important things.  Particularly within the description of the model, it would be good to add some language to help the sections flow- currently they feel very independent.  Alternately, if maybe help if in the beginning part of the model, the different model components (fc, fs, etc.) are named there.  Related, the Architecture Section feels out of place after the Model description.  There are references to the attention layers in the model description which are not explored until the Architecture section.  Perhaps it makes sense to put the Architecture section first because it’s addressing layers/mechanisms that span all aspects of the model.  Or perhaps combining the two sections?  Right now it feels like there are two methods sections.

Some of the text around Eqn(7) seems to be missing because the sentence structure doesn't make sense.

It’s not clear what some of the labels in Figure 3 mean.  You have to go into the text to find out what MLP 4x means, for example, and then when you find it in section 5.2, you have to go back to section 4.3 to actually understand what it means.

The ablation studies feel like they’re done in relative isolation.  It would be useful to know, for example, how the lower performance of using the standard Axial Transformer vs. the conditional Axial transformer impacts the final results, not just that portion.  The section “Conditioning Details” in 5.2 just feels like a results dump.  It’s unclear what motivates those particular ablation choices and what those results tell the reader more generally about this approach.  Some kind of context or discussion would be useful.  In general, this section feels like it’s being included just to show that ablation studies were performed without providing any greater understanding as to the approach (to potentially motivate future work or other examples, for instance).  The descriptions are also very terse.  If these experiments add meaningful insight to this approach, then they belong in the main text with additional explanation and discussion.  If they are merely a justification that this approach works, then I would suggest moving most of this section to the appendix and using the space to give better explanation of the methods and results which are central to the application.

Some of the models which the current method is compared to (Table 2) are not referenced to the best of my knowledge.  What does "CNN" mean in this case?  Do all of these methods use a combined spatial and color upsampling method?  If not, how were they implemented?  This is actually a pretty significant issue as it limits the reproducibility of the comparative experiments.


__3. Recommendation__
Reject.  While the results are compelling, the work largely relies on a method which has not been accepted via peer review.  That in and of itself does not warrant rejection, but I believe it contributed to some of the difficulties in explaining the approach, the motivation behind the approach, the results of the ablation studies, etc., which make the paper extremely difficult to follow, likely difficult to build upon, and potentially difficult to reproduce.


__4. Recommendation Explanation__
I would argue that the main goal of this paper is to show a novel application of the Axial Transformer approach of Ho et al 2019b and this is done by adapting that method to the task of Image Colorization.  I would argue the focus is around applying that method, not exclusively doing better Image Colorization, because there is no discussion around how this advances our understanding of image colorization broadly.  Nevertheless, that (showing the usefulness of an approach to a new task) is a valid objective, but because Ho et al 2019b has not been formally accepted, it also somewhat then requires this work to explain and justify approaches of that work.  I believe that challenge has a lot to do with some of the difficulties in the paper around the methods and experiment explanation.

While the (within sentence) language is clear, the overall flow of the paper is  difficult to follow.  It feels like the authors were strongly up-against the page limit, so important explanation and discussion was omitted or made very terse.  For example, the ablation studies, while thorough, sort of feel dumped there.  There's no discussion as to why those and not other experiments were run and what the results of those experiments tell us more broadly.  Similarly, the model and architecture section seem like they should be more intertwined.  As another example, some of the methods in Table 2 are not referenced anywhere and it's not clear how they were used in this context (did they start with a low res image, or high-res image).  That calls the reproducibility of the comparison studies into question.


__5.  Questions__
Overall it seems like every generated image has a red, green, and blue variant.  Were they sampled in a particular manner to guarantee this?  Obviously it is possible to draw other samples, but do they all largely fall into one of these three coarse categories?  When the performance is poor for a given sample, it usually because entire swaths of the image are being painted in with a very non-natural color (like someone’s face being green, or the entire picture having a blue-ish exposure).  Can you speak to this and other common “mistakes” that are observed?  How do these compare with some of the other methods you compared yours against?  Are there simply fewer “mistakes” (i.e. non-natural images), or are the types of imperfections created by this approach different that would warrant different use-cases?

It seems like a lot of compute (16 TPUv2) was used and the batch size was relatively large.  Is the large batch size necessary for obtaining these results, or could a smaller amount of compute and smaller batch size be used?

Why does training baselines with 2x and 4x wider MLP dimensions make “a fair comparison”?  Is “Baseline” in Figure 3, x1 (standard) MLP but no conditioning?  Why would x1 be better than x4, but worse than x2?

The caption of Figure 2 feels a bit imbalanced.  ColTran core is called out specifically, but then the ColTran Upsamplers are not referenced.  Is the “Axial Transformer” just the right branch of the ColTran Core (which the figure seems to suggest) or the entire ColTran core, as the caption seems to suggest.

On pg. 3 “ColTran Core” it is stated that “we also train a parallel prediction head which we found beneficial for regularization”.  I think it would be useful to given additional explanation here as it’s a fairly significant architectural choice.  If results of not including this head exist, perhaps it would be useful to show this in the appendix.  Otherwise a brief explanation as to why this additional head aids the regularization would be useful.  Since this is an instantiation of the Axial Transformer, is this prediction head added to that approach for this particular task, or is this already a part of the standard Axial Transformer (and therefore maintained here for consistency)?  Ah, this is explored further in section 5.3…. It would be helpful to the reader to reference this section when you introduce the prediction head (i.e. that the impact will be explored in section 5.3).

In 4.2 it says they “adapt the Axial Transformer model for colorization”.  Can you elaborate on the adaptation?  It’s not clear (without looking up that reference), what belongs to the original approach vs. what was added/changed here for this specific task.

It feels odd to mention the number of axial attention blocks in the training section as opposed to the model or architecture.  This is a fundamental architectural choice, is it not?

Why are the set of models compared via FID and Human Evaluation different?


__6.  Feedback__
The demonstrated colorization scores and output are compelling, however, I believe the structure of text is very detrimental.  I think it would potentially be feasible to fully rework the text to make it more readable and reproducible and therefore a solid publication because the result is compelling, but as it stands, there is substantial rewritting which would need to be done in my opinion.

In “Model: ColTran Core” fc is described as a conditional, auto-regressive axial transformer.  While the definition of pc and pc~ are stated thereafter, there is not any further description as to what this means and/or a citation.  The Ho et al. citation is provided in the Figure 2 caption.  At a minimum that citation should be given here as well, but it would be good to give a textual description as to what an “a conditional auto-regressive axial transformer” is since it is not a commonly used architecture.

The second paragraph under ColTran Upsamplers (In our experiments…) is slightly confusing.  It seems to suggest that parallel upsampling is sufficient and advantageous for a number of reasons, but that prediction is chosen to reduce color inconsistencies.  Then it seems to go back to again say that Parallel upsampling has a huge advantage of being fast.  This is perhaps also confusing because there is a “Sample” label in Figure 2.  The confusion is less about the validity of the approach and more that the language (in conjunction with the figure) is difficult to follow for someone not already familiar with Guadarrama 2017.

While not necessary, it would be interesting to see how this approach performs on out of domain images (i.e. not from ImageNet).

In 5.5, it’s stayed you follow the protocol used in PixColor.  It would probably be best to additionally include the citation here, or the citation in place of “PixColor” even though that work is cited near the beginning of the paper (when the reader comes to this section, they may be unfamiliar with this approach and would like to go directly to that reference as opposed to having the search for “PixColor” and then go find the reference).

---

> ### Author Response · Authors · 2020-11-18
> **Review Response: AnonReviewer 4 (Part 1)**
>
> We thank you for investing a significant amount of time in providing detailed reviews and to help us improve the quality of the paper. We have significantly restructured the writing to incorporate your suggestions. We first address the most pressing concerns followed by the minor comments. All our responses are reflected in the latest version of the draft.
>
> **Motivation for using axial transformer**
>
> A summarized version of the points below is added to the end of **Introduction**
>
> * Axial Transformer achieves state-of-the-art on unconditional image generation (at the time of submission) measured using bits-per-pixel on ImageNet32 and Imagenet64 without the usage of custom kernels. It is thus very appealing to use as an autoregressive backbone for colorization.
> * ColTran shares the highly useful advantages of Axial Transformer which is the ability to capture a global receptive field with two layers and efficient implementation using matrix multiplication on modern accelerators such as TPUs.
> * The semi-parallel sampling in Axial Transformer enables us to sample colorizations much faster than prior autoregressive colorization models. As a result, ColTran core can sample 64x64 grayscale images in around 5 minutes P100 GPU vs PixColor that takes ~10 minutes to colorize 28x28 grayscale images on a K40 GPU. Sampling 28x28 colorizations takes just around 30 seconds.
> * AxialDeepLab, a model that applies axial self-attention to semantic segmentation (which at core is the same technique as Axial Transformer modulo masking operations) was recently accepted to ECCV 2020 (https://arxiv.org/abs/2003.07853). We added a citation to this paper in the introduction.
> * The openreview of the Axial Transformer ICLR indicates the method was rejected primarily due to lack of clarity on the contribution of the paper. Two reviewers point out that the claim of the paper was a general purpose technique to improve self-attention in multidimensional transformers while the scope of the paper is indeed limited to autoregressive image modelling.
> * The code for the Axial Transformer is fully open sourced, which can help in removing any ambiguity about the implementation details. We will open source our code as well.
>
> **More explanation on Axial Transformer**
>
> * We expanded the subsection that explains Axial Transformer in section 3.2.
> * The section now contains 3 paragraphs describing the outer decoder, inner decoder and encoder with their corresponding equations.
> * We added a couple of sentences describing the semi-parallel sampling scheme of the Axial Transformer.
>
> **Improving the flow of the paper**
> As requested, we restructured the methods / architecture section in the latest version of the draft and made the following changes. We hope the new narrative is clearer.
>
> * We introduced Section 3 “Background: Axial Transformer” and moved the subsections “Row and Column Self-Attention” and “Axial Transformer” into this. This section is meant for the paper to be self-contained. All terminology revolving the Axial Transformer and axial self-attention is introduced in this section.
> * We combined Section 3 “Model” and Section 4 “Architecture”, to form a “Proposed Architecture” section to make it more intertwined. There are the changes.
>     * **Introduction of Section 4**: Conditional distributions modeled by the three networks.
>     * **Section 4.1**: ColTran core, the equation and the architectural modifications.
>     * **Section 4.2**: Auxiliary parallel head and their equation.
>     * **Section 4.3**: The color and spatial upsamplers and their equations.
> * In short, Section 3 now contains the background material and Section 4 contains the modifications for high-resolution colorization which are contributions of this work.
>
> **Results in Table 2**
> * We added citations to all the baseline models in Table 2 and what underlying generative model they rely on in the Related Work section,
> * We obtained results on FID from cINN [Ardizzone et.al, 2019] and the results on human evaluation results from PixColor. To compute the FID results of PixColor, we used 5000 samples which were provided by the original authors.
> * All of these techniques perform high resolution colorization from a grayscale image, so the numbers are directly comparable. CNN is a deterministic baseline used in [Ardizzone et al 2019]. We removed it as CIC, LRAC and LTBC are also based on deterministic convolutional neural networks.

---

> ### Author Response · Authors · 2020-11-18
> **Review Response: AnonReviewer 4 (Part 2)**
>
> **Ablation studies**
>
> We rewrote the Ablation Studies subsection of the paper. The “conditioning details” section is expanded with bullet points providing a high-level motivation for each experiment. We add it here for convenience.
>
> * We added final FID numbers for the baseline Axial Transformer that conditions only via skip-connections without conditioning layers (**ColTran - B**) in Table 2. The baseline achieves a FID Score of 21.6 (significantly better than the baselines but much worse than ours)
> * **Importance of each conditional component:** We perform a leave-one-out study to determine the importance of each conditional component. We remove each conditional component one at a time and retrain the new ablated model. The curves *no cLN*, *no cMLP* and *no cAtt* in the middle of Figure 3 quantifies our results. While each conditional component improves final performance, cAtt plays the most important role.
> *  **Multiplicative vs Additive Interactions:** Conditional transformer layers employ both conditional shifts and scales consisting of additive and multiplicative interactions, respectively. The curves *Scale* and *Shift* on the right hand side of Figure 3 demonstrate the impact of these interactions via ablated architectures that use conditional shifts and conditional scales only. While both types of interactions are important, multiplicative interactions have a much stronger impact.
> * **Context-aware dot product attention:** Self-attention computes similarity between pixel representations using a dot product between $k$ and $q$, cAtt applies conditional shifts and scales on $q$, $k$ and allow modifying this similarity based on contextual information. The curve *cAtt, only v* depicts that removing this property, by conditioning only on $v$ leads to worse results.
> * **Fixed vs adaptive global representation** cLN aggregates global information with a flexible learnable spatial pooling layer. We experimented with a fixed mean pooling layer forcing all the cLN layers to use the same global representation with the same per-pixel weight. The curve *cLN, mean pool* on the right of Figure 3 shows that enforcing this constraint causes inferior performance as compared to even having no cLN. This indicates that different aggregations of global representations are important for different cLN layers.
> * We expanded the caption in Figure 3 and added a short description of what each label means . This is meant to be a visual aid to the more detailed explanations in Section 5.2
> * We moved the curve computing Gated operations for conditioning and the additional ablation suggested by AnonRev1 to the **appendix G**
>
> **Other questions**
>
> **Q: Overall it seems like every generated image has a red, green, and blue variant. Were they sampled in a particular manner to guarantee this? Obviously it is possible to draw other samples, but do they all largely fall into one of these three coarse categories?**
>
> All our generated images are displayed with pixel-by-pixel sampling. They were not sampled in any other manner. We analyzed what the most dominant coarse color is per-image across 5000 images. The dominant hues ordered by counts are black, white, brown, blue and green. Here is the color band of the top 50 colors (https://ibb.co/Jx9htXq)
>
> **Q: When the performance is poor for a given sample, it usually because entire swaths of the image are being painted in with a very non-natural color (like someone’s face being green, or the entire picture having a blue-ish exposure). Can you speak to this and other common “mistakes” that are observed?**
>
> This is true. Every now and then, we can sample a coarse color for a pixel that has low probability and which the model has not seen before. This can then have a cascading effect leading to such mistakes. Some other mistakes which achieved a 0% fool rate are in Appendix I:
> * Color bleeding when edges are not detected correctly.
> * Inability to color highly complex scenes, such as large no of small objects and complex textures, e.g the dress of a soldier.
> * Once in a while, also we observe that the model returns the grayscale image as a sample. But this is pretty rare.
>
> **Q: How do these compare with some of the other methods you compared yours against? Are there simply fewer “mistakes” (i.e. non-natural images), or are the types of imperfections created by this approach different that would warrant different use-cases?**
> Artifacts such as color-bleeding and unnatural colors are common among the probabilistic colorization models that we compare against on inspection of the samples. You are right that on an average our model generates more natural colorizations avoiding such artifacts given the human evaluation results.
>
> Autoregressive colorization also has a human-in-the-loop use case. For every-pixel, the model can display x most probable colors that the user can choose from and the colorization can be guided by the user.

---

> ### Author Response · Authors · 2020-11-18
> **Review Response: AnonReviewer 4 (Part 3)**
>
> **Q: It seems like a lot of compute (16 TPUv2) was used and the batch size was relatively large. Is the large batch size necessary for obtaining these results, or could a smaller amount of compute and smaller batch size be used?**
>
> 16 TPUv2 chips are the second lowest configuration available to us. As requested, we additionally trained ColTran core and the upsamplers on 4 TPUv2 chips (the lowest configuration) with a reduced-batch size of 56 and 192 each. For the spatial upsampler, we found that a batch-size of 8 was sub-optimal and led to a large deterioration in loss. We thus used a smaller spatial upsampler with 2 axial attention blocks with a batch-size of 16 and trained it also on 4 TPUv2 chips. Our FID drops from 19.71 to 20.9 which is still significantly better than the other models.
> We note that in this experiment, we use just 12 TPUv2 chips in total while PixColor uses a total of 16 GPUs.
> We added the above analysis to **Appendix E**
>
> **Q: Why does training baselines with 2x and 4x wider MLP dimensions make “a fair comparison”? Is “Baseline” in Figure 3, x1 (standard) MLP but no conditioning? Why would x1 be better than x4, but worse than x2?**
>
> We edited the corresponding subsection in **Section 5.2**. Our baselines MLP2x and MLP 4x (now renamed to ColTran-B 2x and ColTran-B 4x) are original Axial Transformer networks that condition via just skip-connections.  Both *ColTran-B 2x* and *ColTran-B 4x* have an increased parameter count via $1 \times 1$ dense layers which are the same operations due to which ColTran has an increased parameter count. So it makes for a fair comparison. Our results show that the increased performance cannot be explained solely by the fact that our model has more parameters.
>
> Re: why x1 performs better than x4, this is purely empirical. Our intuition is that sometimes wider networks can lead to worse performance due to the difficulty in optimization. We ran a small hyperparameter sweep over the learning rates for x1, x2 and x4 and report the best performance.
>
> **Q: The caption of Figure 2 feels a bit imbalanced.**
> We expanded the caption of Figure 2 to give a short description about both ColTran core and the upsamplers. The figure now depicts the "outer decoder", "inner decoder" and "encoder" which were contributions of [Ho et.al 2019]. We now clarify our contributions in the caption itself.
>
> **Q: Auxiliary parallel head**
> This is a contribution of our paper. As noted. we investigate the impact of this auxiliary parallel head in Section 5.3. We added a few words in Section 4.2 that we will study the effect of this later in Section 5.3.
>
> **Q: Adapt the Axial Transformer model for colorization.**
> We removed this. All our architectural modifications or adaptations (i.e the conditional transformer layers and auxiliary parallel head) are now described in Section 4.3 and background information is described in Section 3.
>
> **Q: Number of axial attention blocks**
>
> We believe that this is a hyperparameter of the network similar to the learning rate, optimizer choice and hidden size. We did a very small sweep using the baseline axial transformer (no conditional layers) with the following configurations to come with this number.
> * hidden size = 512, number of blocks = 4
> * hidden size = 1024, number of blocks = 2
> *  hidden size = 512, number of blocks = 2
>
> Once we found the optimal configuration, we fixed this for all future architecture design.
> We added the above to **Appendix F**
>
> **Q: In model, ColTran core….**
>
> In our latest version of the paper, the “ColTran core” paragraph and its corresponding equations have been merged in **Section 4.1 ColTran core**.  We introduce the terminology axial transformer and axial attention in **Section 3** before describing these components.
>
>
> **Q: The second paragraph under ColTran Upsamplers (In our experiments…) is slightly confusing.**
>
> We have moved this explanation to the end of the current section 4.3.
> We upsample all pixels in parallel (parallel upsampling) to predict a distribution over each-pixel in the high resolution image (Eq 6). Instead of sampling from this predicted distribution, we instead use the argmax. There might be a slight confusion between "upsampling" and "sampling" which we clarify.
>
> **Q: Out of domain images might be interesting**
>
> Colorizations of out-of-domain datasets **LSUN-bedrooms** and **Celeb-A** have been added to **Appendix D**. We neither cherry picked these images nor finetune/retrain our model on these datasets. Barring a couple of outliers, our colorizations are realistic. We will colorize “in-the-wild” grayscale images for the final version.
>
> **Q: Probably be best to additionally include the citation here**
> We added the citation of PixColor to this section.
>
> We believe we have clarified all your comments and improved the writing. We are looking forward to reading your updated impression of the paper.

---

### Official Review · AnonReviewer2 · 2020-10-29
**Reasonable approach / Well-written / Better than baselines**

**Rating:** 7
**Confidence:** 4

**Review:**

Update: I really appreciate the authors' efforts to address my original concerns. I believe that this work is a nice application of transformers to image colorization. The paper is well-written and the performance of proposed transformer architecture is strong. I think that this work is above the threshold of acceptance.

**Strengths**
The motivation of the proposed architecture is reasonable. The paper is generally well-written.

**Major comments**
It’s better to include some discussion on regularization effects from Eq. (4). Eq. (4) seems to be helpful to capture the overall structure in an image, rather than capturing only local correlation from autoregressive formulation.

For upsampling, do we really need to make use of an autoregressive model? A stack of transposed convolutions might be working well, because we only need to upsample input/color resolutions. I totally agree that autoregressive formulation does help achieve better results, but it may be possible to achieve similar performance by just using transposed convolutions.

It’s better to include more details for reproducing results.
(1) Even if different batch sizes used (224, 768 and 32), learning rates for all experiments are fixed 3e-4?
(2) How many epochs or steps are required for convergence?
(3) Figure 6 shows that EMA is extremely important. How about using cosine annealing for a learning rate scheduler? It may help achieve more robust FID scores without EMA.
(4) Compared to baselines, this approach is extremely slow due to the autoregressive sampling. It’s better to report inference time.

I'm not sure that conditional layer normalization is indeed helpful.

**Minor comments**
The x-axis title of figure 4c (“training steps”) seems to be wrong.

---

> ### Author Response · Authors · 2020-11-18
> **Review Response: AnonReviewer 2**
>
> We thank you for your reviews. Please find attached our response.
>
> **Effect of the auxiliary parallel model**
> We added a bit that this helps to capture global structure in Section 4.2. We perform a detailed empirical analysis on the effect of this model in **Section 5.3**
>
> **Upsamplers**
> We would like to clarify that our upsampling is done by parallel self-attention based models and not autoregressive. However, it is true that upsampling / refinement can be potentially done by convolutional architectures. However:
> * The spatial refinement network in PixColor uses 28M parameter whereas our spatial upsampler uses just 13M parameters. We require deeper convolutional networks to perform upsampling.
> * From a practical perspective, it makes our architecture a bit more complicated. Currently our architecture is conceptually simple and employs only axial attention blocks with optional / conditioning + masking. In future, we could explore combining convolutions + attention in different parts of the network to improve colorization performance.
>
> **Other**
> * Yes, that is true, we used the same learning rate for all models. The spatial upsampler receives gradients (albeit correlated) from 256x256 pixels, as compared to the colorizer and the color upsampler, which might explain why a smaller batch-size for the spatial upsampler is sufficient. We did not tune hyperparameters extensively. Once we found an architecture that colorizes low resolution images coarsely, we used the same training and architecture setup for the color and spatial upsampler. It may be possible to improve our results with an extensive hyperparameter sweep.
> * We train our colorizer for 450K steps, the color upsampler for 300K steps and spatial upsampler for 150K steps. In general, we found longer training to improve performance for the colorizer and not so much for the upsamplers. We added this to the training subsection in **Section 5.1**
> * Applying a cosine learning rate schedule requires 2 hyperparameters to tune whereas we apply polyak averaging with a value of 0.999, In future, we can experiment with different learning rate schedules.
> * ColTran core can sample 64x64 grayscale images in 4-5 minutes P100 GPU vs PixColor that takes ~10 minutes to colorize 28x28 grayscale images on a K40 GPU. Sampling 28x28 colorizations takes just around 30 seconds. The upsampler networks take in the order of milliseconds. We added this analysis to **Appendix H**.

---

### Official Review · AnonReviewer3 · 2020-10-29
**Lack of clarity and novelty, weak evaluation**

**Rating:** 5
**Confidence:** 4

**Review:**

Thank the authors for addressing reviewers' comments extensively. After rebuttal, I agree with the significance of the proposed method in terms of performance improvement in this particular task. However, the technical novelty is still limited. Thus, I increased my rating to 5.

In this paper, the authors propose an autoregressive image colorization method based on self-attention. The proposed method first infers an initial low-resolution colorization in an autoregressive manner, then upsamples both spatial resolution and color depth. The authors adopt self-attention to encode contextual information of the scene. Experimental results show that each component of the proposed method is effective and the proposed method outperforms an existing autoregressive method.

Overall, it is difficult to understand the contribution of this paper. I think it is because the writing in Sec. 1 and 2 is unclear. Particularly, the writing of introduction needs a significant improvement as the authors reveal too much details of this paper instead of describing the high-level motivation of the proposed method and the technical contribution. The clarity also needs to be improved in the method and experiment sections. (e.g. ColTran Core in Fig. 2 is confusing. It looks complicated, but the writing is too short. what is the ground-truth in the objective? what about Table 2? each baseline is not explained and cited.)

Technical novelty is incremental. I could not understand the motivation of the proposed network due to the clarity issue, but this paper generally adopts existing methods such as an autoregressive model and self-attention blocks to apply them to an image colorization problem, which limits the novelty of this paper.

Evaluation is weak. PixColor is an old model (in 2017), so recent methods and state-of-the-art methods should be compared. I could not find out what the baseline methods in Table 2 are, but they do not look like state-of-the-art models. Performance gain over the previous autoregressive model using widely-used self-attention blocks is not enough for accepting this paper.

---

> ### Author Response · Authors · 2020-11-18
> **Review Response: AnonReviewer 3 [1/2]**
>
> We thank you for your reviews. Please find attached our response. We believe the latest version of the draft improves clarity, highlights the contributions of our work and motivates the architecture better.
>
> **Technical contributions**
> We have added a summarized version of the following points to the end of the **Introduction (Section 1)** highlighting the technical contributions of our paper.
> * First application of transformers for high-resolution ($256 \times 256$) image colorization. Axial transformers which we base our technique upon were initially applied to model only low resolution $64 \times 64$ images. Other related techniques that rely exclusively on self-attention to model images such as the Sparse Transformer [Child et. al 2019] and Image Transformer [Parmar et. al 2019] limit to resolutions of 64x64 and below. Scaling transformers to the task of coloring $256 \times 256$ grayscale images or equivalently modeling $\sim$ 200K symbols is a challenging task and our paper accomplishes this task quite successfully.
> * While Axial Transformers support conditioning by biasing the input, we find that **directly conditioning the transformer layers** can improve results significantly. We introduce conditional transformer layers for low-resolution coarse colorization in **Section 4.1**. The conditional layers incorporate conditioning information via multiple learnable components that are applied per-pixel and per-channel. We validate the contribution of each component with extensive experimentation and ablation studies that were appreciated by multiple reviewers. Our experiments can provide insight on how to effectively condition spatial information in a transformer for related tasks such as image editing and restoration.
> * We propose training an **auxiliary parallel prediction model** jointly with the low resolution coarse colorization model in **Section 4.2**. Improved FID scores demonstrate the usefulness of this auxiliary model.
>
> **Motivation**
> We agree that the motivation was not clearly described in the first draft of our paper. Here we describe the motivation behind different components of our architecture. We added a summarized version discussing this towards the end of the **third paragraph in Section 1**
> *  **Motivation for axial self-attention blocks**  -  The main advantages of axial self-attention blocks are the ability to capture a global receptive field with only two layers and $\mathcal{O}(D \sqrt{D})$ instead of $\mathcal{O}(D^2)$ complexity. They can be implemented efficiently using matrix-multiplications on modern accelerators such as TPUs.
> * **Motivation for using three sub-networks**: Generating high-resolution images using only self-attention is computationally challenging and hence prior work on unconditional image generation using self-attention limits to generating small images $64 \times 64$. To alleviate the inherent complexity in colorizing high-resolution grayscale images, we decompose the task into three simpler sequential subtasks: coarse low resolution colorization, color super-resolution and spatial super-resolution and use a separate network for each. This enables us to train larger models for colorization.
> * **Motivation for the choice of Axial Transformer** - Axial Transformer is state-of-the-art in unconditional image generation benchmarks (ImageNet 32, ImageNet 64) at the time of submission without the usage of custom GPU kernels. The Axial Transformer has a semi-parallel sampling mechanism which enables us to colorize 64x64 grayscale images in 4-5 minutes P100 GPU vs PixColor that takes ~10 minutes to colorize 28x28 grayscale images on a K40 GPU. Sampling 28x28 colorizations takes just around 30 seconds.
> * **Motivation for conditional transformer layers** - Conditioning every layer via multiple components allows stronger gradient signals through the encoder and as an effect the encoder can learn better contextual representations. This improves our results compared to a baseline Axial Tranformer both in Table 2 and Figure 3.
>
> **Clarity**
> In the latest version of the draft, we have restructured the writing to improve readability.
> * We added citations to Table 2. We provide a short description of what generative model each baseline is based upon in the Related Work section.
> * We expanded the caption in Figure 2. We expanded the "Axial Transformer" subsection and now explain every component of the figure (Outer Decoder, Inner Decoder, Encoder) in detail with equations. Our modifications made to the architecture are now described in Section 4.
> * The ground-truth coarse low resolution image is both the input to the decoder and the target during training. Masked layers ensure that the conditional distribution over each pixel depends solely on information from previous ground-truth pixels.

---

> ### Author Response · Authors · 2020-11-18
> **Review Response: AnonReviewer 3 [2 / 2]**
>
> **Evaluation**
>
> * We added the most recent baseline in the colorization literature cINN, [Ardizzone et al, 2019], that uses a combination of a VGG network + Glow [Kingma et al, 2018] in Table 2. The model performs slightly worse than PixColor.
> * We extend the original axial transformer [Ho 2019 et al] for image colorization which is a contribution of our work. This is a strong baseline in itself. (ColTran - B) in Table 2 and Figure 3.
> * We improve upon this baseline significantly by our architectural modifications. Comparisons to ColTran-B are provided in Table 2 and Figure 3.
> * We did an extensive literature survey on existing generative colorization techniques in Section 2 as noted by.Reviewer 1. PixColor, despite coming out in 2017, is still a state-of-the-art colorization model.
> * Image colorization is a underexplored yet important research area (noted by Reviewer 1 and 4). Hence it is not entirely surprising that there is a lot of scope to improve existing state-of-the-art colorization techniques as compared to unconditional image generation.
> * The quality of our colorizations are almost imperceptible from the ground-truth barring a few outliers as reflected in mechanical turk results.
> * All-in-all,  we believe advancing the state-of-the-art significantly (~20% relative improvement) over prior techniques which all have FID scores between 24 and 26 and setting a strong baseline for future research in colorization should be considered a positive of the paper and not a negative.
>
> We believe that our latest version of the draft plus our response clarifies some of the concerns raised.

---

### Author Response · Authors · 2020-11-18
**Rebuttal**

Thanks everyone for your time and reviews. Here are a summary of the changes

**Writing**
* Restructured the end of the introduction. It now highlights the motivation of the network and contributions of the paper.
* Moved **Row and Column Self Attention** and **Axial Transformer** to **Section 3 Background: Axial Transformer**
* Expanded *subsection “Axial Transformer” with paragraphs **Outer Decoder, Inner Decoder and Encoder** and equations.
* Merged the remainder of the “Model” section and “Architecture” into a **Section 4 Proposed Architecture**
* Expanded the **Ablation Studies (Section 5.2)** to add some insights from each experiment
* Added citations to all models in **Table 2**
* Added a small background on autoregressive models to **Appendix A**
* Added #parameters and inference speed comparisons to the **Appendix H**
 * Edit: Nov 20, provided a bit more detail on the semi-parallel sampling mechanism.

**Experiments**
* Added out-of-domain colorizations on Celeb-A and LSUN **Appendix D**
* Added experiments on the low-batch size regime **Appendix E**
* Added final FID results on the baseline Axial Transformer to **Table 2 (ColTran-B)**
* Added shift modulation experiments to the graph in **Figure 3**.
* Added global conditioning on cAtt and cMLP experiment **Appendix G**

Given the strong empirical performance and experimentation (mentioned by multiple reviewers) and our now improved writing,
We hope that all our contributions as a whole would be of interest to the ICLR audience.

Please reconsider your scores after the rebuttal revision.

---

### Author Response · Authors · 2020-11-24
**End of discussion phase**

Hi all,
Since the end of the discussion phase is fast approaching, we would like to know if we our rebuttal helped to clarify some concerns. If there were other concerns that we could help to clarify, please do let us know.
Thanks.

---

### Decision · Program_Chairs · 2021-01-07
**Final Decision**

**Decision:**

Accept (Poster)

**Comment:**

The paper initially received a mixed rating, with two reviewers rate the paper below the bar and two above the bar. The raised concerns include the need for an autoregressive model for upsampling and the effect of batch sizes. These concerns were well-addressed in the rebuttal. Both of the reviewers that originally rated the paper below the bar raise the scores. After consulting the paper, the reviews, and the rebuttal, the AC agrees that the paper has its merits and is happy to accept the paper.